# Investigating the sensitivity to resolving aerosol interactions in downscaling regional model experiments with WRFv3.8.1 over Europe

Vasileios Pavlidis[1], Eleni Katragkou[1], Andreas Prein[2], Aristeidis K. Georgoulias[1], Stergios Kartsios[1], Prodromos Zanis[1], and Theodoros Karacostas[1]

[1]Department of Meteorology and Climatology, School of Geology, Aristotle University of Thessaloniki, Thessaloniki, Greece
[2]National Center for Atmospheric Research, Boulder, CO, USA

**Correspondence:** Vasileios Pavlidis (vapavlid@physics.auth.gr)

**Abstract.**

In this work we present downscaling experiments with the Weather Research and Forecasting model (WRF) to test the sensitivity to resolving aerosol-radiation and aerosol-cloud interactions on simulated regional climate for the EURO-CORDEX domain. The sensitivities mainly focus on the aerosol-radiation interactions (direct and semi-direct effects) with 4 different aerosol optical depth datasets (Tegen, MAC-v1, MACC, GOCART) being used and changes to the aerosol absorptivity (single scattering albedo) being examined. Moreover, part of the sensitivities also investigates aerosol-cloud interactions (indirect effect). Simulations have a resolution of 0.44º and are forced by the ERA-Interim reanalysis. A basic evaluation is performed in the context of seasonal-mean comparisons to ground based (E-OBS) and satellite-based (CMSAF SARAH, CLARA) benchmark observational datasets. The impact of aerosol is calculated by comparing against a simulation that has no aerosol effects. Implementation of aerosol-radiation interactions reduces the direct component of the incoming surface solar radiation by 20-30% in all seasons, due to enhanced aerosol scattering and absorption. Moreover the aerosol-radiation interactions increase the diffuse component of surface solar radiation in both summer (30-40% ) and winter (5-8% ) whereas the overall downward solar radiation at the surface is attenuated by 3-8% . The resulting aerosol radiative effect is negative and is comprised of the net effect from the combination of the highly negative direct aerosol effect (-17 to –5 W/m$^2$) and the small positive changes in the cloud radiative effect (+5 W/m$^2$), attributed to the semi-direct effect. The aerosol radiative effect is also stronger in summer (-12 W/m$^2$) than in winter (-2 W/m$^2$). We also show that modelling aerosol-radiation and aerosol-cloud interactions can lead to small changes in cloudiness, mainly regarding low-level clouds, and circulation anomalies in the lower and mid-troposphere, which in some cases, mainly close to the Black Sea in autumn, can be of statistical significance. Precipitation is not affected in a consistent pattern throughout the year by the aerosol implementation and changes do not exceed $\pm$ 5% except for the case of unrealistically absorbing aerosol. Temperature, on the other hand, systematically decreases by -0.1 to -0.5ºC due to aerosol-radiation interactions with regional changes that can be up to -1.5ºC.

# 1 Introduction

Aerosols play an important role in the Earth's climate system due to their substantial effects on the radiation budget and cloud properties (Ramanathan et al., 2001). The 5[th] Climate assessment report of the Intergovernmental Panel on Climate Change (IPCC) (Boucher et al., 2013) identifies aerosols together with clouds as the largest sources of uncertainty in the Earth's climate system. It states that the uncertainty due to aerosol is attributed to both aerosol-radiation (ari) and aerosol-cloud interactions (aci) with the latter having the largest contribution. In the regional climate model experiments of the Coordinated Regional Climate Experiment (CORDEX) (Giorgi and Gutowski, 2015) covering the European and Mediterranean regions (EURO-CORDEX, MED-CORDEX), aerosols are treated differently in the various participating modelling systems. Within the MED-CORDEX community there have been several studies highlighting the impacts of aerosols (Ruti et al., 2016). The considerable impact of the aerosol direct and semi-direct effect (also known as aerosol-cloud semi direct effect; Allen et al.,2019) on the climate of the Euro-Mediterranean region has been clearly demonstrated (Huszar et al., 2012; Zanis, 2009; Zanis et al., 2012; Nabat et al., 2015). The substantial impact of centrain aerosol species, such as the African dust, to the greater region has also been established (Tsikerdekis et al., 2019). Moreover, long-term trends in aerosol concentrations have been linked to observed trends in temperature and radiation over the Euro-Mediterranean region (Nabat et al., 2014) that cannot be reproduced without considering aerosol effects in RCM simulations. Inclusion of aerosol representation is also considered essential in solar energy generation (Gutiérrez et al., 2018). Within EURO-CORDEX co-ordinated experiment the treatment of aerosol depends on the modelling system and on the model setup[1] : the majority of the models participating in the experiment takes aerosols into account by using aerosol climatologies either in a time-invariant manner or with monthly variations that partly include trends while a few models do not include aerosols at all. Finally only one model uses a prognostic aerosol scheme estimating online the aerosol field. The aerosol climatologies, used by the majority of the models, are not consistent and some models use outdated datasets. In a modeling study over Europe, Zubler et al. (2011) has shown that changing to newer aerosol climatologies can have a significant impact on model results, specifically on shortwave radiation at the surface. Schultze and Rockel (2018) have also shown improvement of model performance when using newer aerosol climatologies on long-term climate simulations over Europe. The Weather Research and Forecasting (WRF) model (Powers et al., 2017; Skamarock et al., 2008) has previously been used to explore the impact of aerosol on weather and climate patterns. Ruiz-Arias et al. (2014) introduced an aerosol-radiation interaction parameterization and tested it over the continental U.S. to investigate its impact on radiation. They concluded that the parameterization produces satisfactory results for predicting shortwave radiation at the surface and its direct and diffuse components. Moreover they demonstrated that the inclusion of aerosol-radiation interactions significantly reduces prediction errors in radiation under clear sky conditions, especially in simulating diffuse radiation. Furthermore the seasonality of the radiation bias is also improved when the seasonal variability of the aerosol optical depth is taken into account. Similar results were documented by Jimenez et al. (2016) by implementing aerosol-cloud-radiation feedbacks into WRF with the use of the new Thompson aerosol-cloud interacting (aerosol-aware) cloud microphysics scheme (Thompson and Eidhammer, 2014) that is computationally inexpensive enough to support operational weather and solar forecasting. This aerosol-cloud interaction

---

[1]https://docs.google.com/document/d/1UCCv-DU8hLlZaSPkcndnM0SrJHoX4cvG-yqxbIDZlRc/edit

option is available from WRF v3.6 onward. Da Silva et al. (2018) used this aerosol-cloud interacting cloud microphysics scheme in WRF to estimate the aerosol indirect effect and its impact on summer precipitation over the Euro-Mediterranean region, concluding that higher aerosol loads lead to decreased precipitation amounts. Here we use the WRFv3.8.1 model, which is widely used for regional climate simulations over Europe (Katragkou et al., 2015). The scope of this paper is to first

evaluate the AOD of the datasets used (section 3.1) and the model simulation without aerosol treatment (section 3.2) and then examine the impact of aerosol radiation interactions on the European climate, including different aerosol parameterizations and model configurations as well as aerosol climatologies (section 3.3). In section 3.4 we present the impact of aerosol-radiation interactions when the aerosol-cloud interactions are also enabled. Finally, in section 3.5 we assess the impact of the Thompson aerosol-cloud interacting microphysics scheme. We examine various radiation components, which are commonly not examined

in RCM simulations (total, clear sky, direct, diffuse radiation), clouds, temperature and precipitation.

## 2   Data and Methodology

### 2.1   Observational Data

#### 2.1.1   Temperature and precipitation

The evaluation of the model simulations for temperature (2m) and precipitation is performed against the E-OBS v16 dataset

(Haylock et al., 2008). Daily mean values are used covering Europe on a 0.44º rotated pole grid. It is a gridded dataset with good spatial and temporal coverage, however, as with all datasets, it is not without limitations. When compared against regional datasets with higher station density (Hofstra et al., 2009) the E-OBS dataset presented a mean absolute error around 0.5ºC for temperature whereas for precipitation a general tendency of underestimating precipitation amount is reported, with large (>75%) relative errors found in mountainous regions of the Alps and Norway, over North Africa and in areas east to the

Baltic Sea. Moreover, Prein and Gobiet (2017) showed that uncertainties in European gridded precipitation observations are particularly large in mountainous regions and snow dominated environments.

#### 2.1.2   Radiation

Shortwave downwelling radiation flux at the surface (Rsds) and Direct Normalized Irradiance at the surface (DNI) are compared against the Surface Solar Radiation Data Set - Heliosat (SARAH)-Edition1 (Müller et al., 2015). DNI is the solar radiation

received by the direction of the sun's rays and received by a surface that is perpendicular to that direction. The SARAH dataset is based on satellite observations coming from the MVIRI and SEVIRI instruments onboard the geostationary Meteosat satellites. SARAH is available as hourly, daily and monthly averages on a regular grid with a high spatial resolution of $0.05° \times 0.05°$ from 1983 to 2013 between $\pm 65°$ longitude and $\pm 65°$ latitude. Here we use monthly values. Another satellite product used for Rsds evaluation in this study is the CLARA-A1 dataset (Karlsson et al., 2013). This is a global dataset which contains a number

of cloud, surface albedo and surface radiation products. In contrast to the SARAH dataset, CLARA is based on observations from polar orbiting NOAA and Metop satellites carrying the Advanced Very High Resolution Radiometer (AVHRR). It covers

the period from 1982 to 2009 globally on a regular 0.25 degree spacing latitude-longitude grid. Both SARAH and CLARA-A1 satellite datasets were obtained from CMSAF (Satellite Application Facilities for Climate Monitoring), which is part of the European Organization for the Exploitation of Meteorological Satellites (EUMETSAT). SARAH has less missing values, better accuracy ($< 5W/m^2$) and less estimated uncertainty ($<10W/m^2$) for Rsds compared to the CLARA dataset (Karlsson et al., 2013; Müller et al., 2015). According to our analysis discrepancies between the two datasets do not generally exceed 15% for most subregions and seasons. Larger differences can be found in Scandinavia during winter, possibly related to its high latitude, which can be challenging for geostationary satellites as those used in SARAH (Schulz et al., 2009), and to the high albedo due to extensive snow coverage. Since relative differences between the two sets are small, and spatial correlation is quite high (0.95 to 0.98 depending on season) we only use the Rsds observations from the SARAH dataset for model evaluation.

### 2.1.3 Cloud fraction

Here cloud fraction means total column cloud fraction. Our primary source of cloud fraction data is the CLARA-A1 satellite dataset described above (section 2.1.2). In an evaluation (Karlsson and Hollmann, 2012) against global synoptic cloud observations (for the period 1982-2009) the CLARA cloud fraction product has shown a small overestimation of 3.6% whereas against satellite-based observations from the CALIOP/ CALIPSO instrument (for the period 2006-2009) it exhibited and underestimation of -10% . The use of a different product for cloud fraction (CLARA) than the one used for radiation (SARAH) does not impact the evaluation since both of these products have reasonable accuracy and uncertainty in estimating the respective variables.

### 2.1.4 Aerosol optical depth

In order to assess the aerosol data used in our simulations (section 3.1) we use the aerosol optical depth (AOD) at 550nm of the MODIS Level-3 (L3) Atmosphere Monthly Global Product (Platnick, 2015; Hubanks et al., 2019). This is a satellite gridded dataset of various atmospheric parameters having global coverage on a 1x1 degree resolution. It monitors AOD for non cloudy conditions in daytime. We use monthly mean values of $AOD_{550}$. To increase robustness we also use $AOD_{550}$ estimates of the CMSAF climate data record (Clerbaux et al., 2017). This dataset is derived from measurement of the SEVIRI instrument, on the Meteosat Second Generation satellite, after the incorporation of the Land Daily Aerosol (LDA) algorithm. Monthly $AOD_{550}$ estimates have been used for this study.

### 2.2 Model

All simulations in this work are performed with WRF/ARW (version 3.8.1) model (Skamarock et al., 2008; Powers et al., 2017). The domain covers Europe (25N-75N, 40W-75E) with a resolution of 0.44º ($\sim$ 50km) following the EURO-CORDEX specifications (Giorgi and Gutowski, 2015) and domain setup. The simulations are forced by the ERA-Interim reanalysis (Dee et al., 2011) while the same dataset is used for the imposed sea surface temperature (SST) variations. The model has 133X130 grid points and 31 vertical levels reaching up to 50 hPa with a 9 grid cells relaxation zone at the model top. The selected

time period for the sensitivity study extends from 2004 to 2008 (2003 used as spin up time) to allow for comparison with the EUMETSAT satellite datasets. All simulations are conducted with the same model setup and parameterizations with the only differences being the aerosol options and aerosol data used (see details in section 2.4).

In our regional climate modeling sensitivity experiments, we use the Thompson cloud microphysics scheme (Thompson et al., 2008) in six simulations and the Thompson aerosol-cloud interacting cloud microphysics scheme (Thompson and Eidhammer, 2014) in two simulations (Table 1). The aerosol-cloud interacting scheme is based on the Thompson bulk scheme, which is double moment regarding cloud ice and rain and uses five hydrometeor species: cloud water, cloud ice, rain, snow and graupel. The aerosol-cloud interacting scheme incorporates aerosols in the microphysical processes, thus enabling aerosol-cloud interactions (indirect aerosol effect) which are absent in the previous Thompson (2008) cloud microphysics scheme.

All simulations use the land surface model CLM4 (Lawrence et al., 2011; Oleson et al., 2010), the planetary boundary layer scheme from the Yonsei University (Hong et al., 2006), the revised-MM5 surface layer option (Jiménez et al., 2012) and the Grell-Freitas cumulus scheme (Grell and Freitas, 2014). The RRTMG (Iacono et al., 2008) radiation scheme is used to simulate short and longwave radiation, which is compatible with the aerosol-radiation interaction implementation in the aerosol-cloud interacting Thompson cloud microphysics scheme. Model cloud fraction has been calculated using the method described in (Sundqvist et al., 1989) (icloud=3 option in the namelist). This is based on a threshold of relative humidity (RH) which is affected by the grid size. The "cu_rad_feedback" flag is also enabled to allow sub-grid cloud fraction interaction with radiation (Alapaty et al., 2012).

## 2.3 WRF Aerosol options and input data

### 2.3.1 WRF aerosol parameterizations examined

*Aerosol-radiation interactions*

All the aerosol-radiation parameterizations examined regard the RRTMG radiation scheme. The WRF model provides three main aerosol options encompassing aerosol-radiation interactions for the RRTMG scheme. The first, (aer_opt=1 in the namelist) uses the aerosol input climatology of Tegen et al. (1997). The spatial resolution of the data is coarse (5 degrees in longitude and 4 degrees in latitude) and temporal changes throughout the year are included as monthly variations. For its implementation in WRF, AOD is provided in each vertical model level, as an aggregate of the five aerosol types taken into account (organic carbon, black carbon, sulfate, sea salt and dust). The single scattering albedo (SSA) and asymmetry factor (ASY) are given for each aerosol type and a final value is calculated in each model level and for each spectral band of the radiation scheme. This is done by weighting the value of each aerosol type by its respective AOD and aggregating for all five aerosol types. SSA values range from 0.85 over North Africa to 0.98 over the Atlantic with typical values over continental Europe being around 0.9.

The second aerosol-radiation option (aer_opt=2) (Ruiz-Arias et al., 2014) enables the user to provide aerosol input data. The user can either provide non-variable aerosol properties in the namelist or an external aerosol data file with spatial and temporal aerosol variations. In the latter option, the user must provide the total column aerosol optical depth at 550nm (AOD550) and can either choose to provide other aerosol optical parameters ( single scattering albedo (SSA), the asymmetry factor (ASY)

and Angstrom exponent (AE) ) or can choose to parameterize one or all of them through selecting a certain "aerosol type" in the namelist. There are three aerosol types available, rural, urban and maritime. In this work we use the first two options. The "rural" option considers aerosols as a mixture of 70% water soluble and 30% dust aerosols. The "urban" type consists of 80% of the above "rural" type aerosols mixed with 20% soot aerosols, thus making it considerably more absorbing. Only one aerosol "type" can be used for the entire domain. Finally, the vertical distribution of aerosol AOD is described with a prescribed exponential profile. This is adequate for assessing the impact of total aerosol load on the radiation at the surface, but studying aerosol-radiation interactions at vertical levels (possible semi-direct effect) could possibly be incomplete with this assumption. Using the second aerosol option (aer_opt=2) we conducted simulations with two aerosol datasets. The third aerosol option (aer_opt=3) enables aerosols to interact with radiation within the Thompson aerosol-cloud interacting cloud microphysics scheme. It is based on the second aerosol-radiation option described above using the "rural" aerosol type. Further information about the aerosol of the new Thompson aerosol-cloud interacting cloud microphysics can be found in the next paragraph 2.3.2

The third aerosol option (aer_opt=3) enables aerosols to interact with radiation within the Thompson aerosol-cloud interacting cloud microphysics scheme. It is based on the second aerosol-radiation option described above using the "rural" aerosol type. Further information about the aerosol of the new Thompson aerosol-cloud interacting cloud microphysics can be found in the next paragraph 2.3.2 Aerosol options one and three can only be used with the RRTMG radiation scheme whereas option two can also be used with the Goddard radiation scheme.

*Aerosol-cloud interactions*

The new Thompson aerosol-cloud interacting cloud microphysics scheme has an internal treatment of aerosols. Aerosols are separated into cloud droplet nucleating acting as cloud condensation nuclei (CCNs), and cloud-ice nucleating, acting as ice nuclei (IN). Cloud-droplet nucleating aerosols include sulfates, sea salt and organic carbon. Cloud-ice nucleating aerosols include dust larger than $0.5 \mu$ m. Black carbon is not included. This scheme explicitly predicts aerosol number concentrations. Aerosol initialization and boundary conditions are based on an aerosol climatology constructed from global simulations spanning the period 2001-2007 (Colarco et al., 2010) with the use of the Goddard Chemistry Aerosol Radiation and Transport (GOCART) model (Ginoux et al., 2001).The two categories of aerosols are then advected and diffused during the model run. Furthermore, a field representing cloud-droplet nucleating surface aerosol emission flux is introduced to the lowest model level at each time step. Surface emission flux is based on initial aerosol concentrations at the surface and on a constant value of mean surface wind. Aerosols are free to either change cloud albedo (first indirect or Twomey effect) or/and impact cloud lifetime (second or Albrecht indirect effect). Moreover, aerosols can be allowed to interact with radiation (aer_opt=3), enabling aerosol-radiation interactions in addition to the existing aerosol-cloud interactions, thus providing a complete representation of aerosol interactions.

### 2.3.2 Aerosol datasets used

We use two external aerosol datasets. The first is the Max-Planck-Institute Aerosol Climatology version 1 (MAC-v1) (Kinne et al., 2013). The MAC-v1 is a global climatology of aerosol that has been produced by combining global aerosol models and ground-based measurement by sun-photometer networks. Aerosol optical properties are provided on a global scale at a spatial

resolution of 1 degree. Monthly data regarding total, as well as anthropogenic aerosol properties, are available ranging from preindustrial times to the end of 21$^{st}$ century. We use a part of this climatology that contains the merging of monthly statistics of aerosol optical properties to describe current conditions.

The second dataset used is the MACC reanalysis (Inness et al., 2013). Data are provided globally at a horizontal resolution of about 80 km for the troposphere and the stratosphere. An advantage of the MACC dataset is its daily resolution. A study that tested different climatologies (Mueller and Träger-Chatterjee, 2014), including MAC-v1 and a climatology based on the MACC reanalysis concluded that the MACC climatology leads to the highest accuracy in solar radiation assessments.

## 2.4   Model Simulations

Using the above aerosol options and datasets we performed 7 sensitivity experiments from a control run with no aerosol interactions covering the period 2004-2008.

- The control experiment (CON) does not include aerosol-radiation or aerosol-cloud interactions (aer_opt=0), meaning the simulation is aerosol-insensitive.

- The second simulation including aerosol-radiation interactions (ARI_T) uses the Tegen (1997) climatology (aer_opt=1).

The next four experiments, also only account for aerosol-radiation interactions and use the methodology introduced by Ruiz-Arias, Dudhia, and Gueymard (2014) (aer_opt=2):

- ARI_Mv1 uses AOD$_{550}$ from the MAC-v1 climatology and the "rural" aerosol type.

- ARI_Mv1urban uses AOD$_{550}$ from the MAC-v1 climatology as well but assigns all aerosols to the more absorbing "urban" aerosol type.

- ARI_Mv1full uses AOD$_{550}$, single scattering albedo (SSA) and asymmetry factors (ASY) at 550nm from the MACv1 climatology together with the "rural" aerosol type to parameterize only the Angstrom exponent (AE).

- ARI_MC uses the MACC aerosol optical depth at 550nm dataset and the "rural" aerosol type.

All of these simulations use the Thompson (mp=8) aerosol-cloud interacting cloud microphysics scheme which will be referred to as the Thompson2008 scheme. It must be noted here that implementation of aerosol-radiation interactions in a simulation enables the impact of both the direct and the semi-direct aerosol effect. The single scattering albedo (SSA) at 550nm of the "rural" type aerosols ranges in our experiments between 0.92 and 0.98 whereas the "urban" type is much more absorbing with SSA starting as low as 0.6, values that are considered unrealistic (Rodriguez et al., 2013; Tombette et al., 2008; Witte et al., 2011). Therefore the ARI_Mv1urban simulation must be considered as an idealized experiment of extremely absorbing aerosols.

Two additional simulations (ACI, ARCI) have been performed using the new Thompson aerosol-cloud interacting cloud microphysics scheme (mp=28 in the namelist), which enables the aerosol indirect effect (aerosol-cloud interactions).

- The ACI simulation does not consider aerosol-radiation interactions.

- Simulation ARCI includes both aerosol-radiation and aerosol-cloud interactions. This simulation presents the most complete physical description of aerosol effects in the simulation ensemble.

All the simulations, aerosol sources and options used are presented in Table 1. The simulations that account for aerosol-radiation interactions are symbolized with ARI in their names. Within the ARI group simulations ARI_Mv1, ARI_Mv1urban and ARI_Mv1full have the same $AOD_{550}$ field (MAC-v1) but they have differences in the rest aerosol optical properties (single scattering albedo, asymmetry factor). The simulation with the Thompson aerosol-cloud interacting scheme that accounts for aerosol-cloud interactions is symbolized as ACI whereas the experiment that accounts both for aerosol-radiation and aerosol-cloud interactions is symbolized as ARCI. The simulations that account only for aerosol-radiation interactions will be referred to as the ARI group of experiments. Finally, for brevity, the Thompson aerosol-cloud interacting scheme is referred to as TE2014 hereafter.

**Table 1.** Simulations conducted and description of aerosol treatment

| Simulation | CON (Control) | ARI_T | ARI_Mv1 | ARI_Mv1urban | ARI_Mv1full | ARI_MC | ACI | ARCI |
|---|---|---|---|---|---|---|---|---|
| **Cloud micro-physics scheme** | Thompson 2008 | Thompson 2008 | Thompson 2008 | Thompson 2008 | Thompson 2008 | Thompson 2008 | TE2014 | TE2014 |
| **Aerosol option** | - | aer_opt=1 | aer_opt=2 | aer_opt=2 | aer_opt=2 | aer_opt=2 | aer_opt=0 | aer_opt=3 |
| **Aerosol source** | - | Tegen | MAC-v1 | MAC-v1 | MAC-v1 | MACC | GOCART | GOCART |
| **User in-put data** | - | No input by user | AOD, "rural" aerosol type | AOD, "urban"" aerosol type | AOD,SSA, ASY "rural" aerosol type | AOD, "rural" aerosol type | - | - |
| **Aerosol inter-acting with** | - | **radiation** | **radiation** | **radiation** | **radiation** | **radiation** | **clouds** | **radiation + clouds** |

## 2.5 Methodology

We analyze the following variables: temperature at 2m, precipitation, shortwave downwelling radiation at the surface (Rsds), direct normalized irradiance at the surface (DNI), diffuse irradiance at the surface (DIF), total cloud fraction (CFRACT) and the wind field at various pressure levels. Direct normalized irradiance is the solar radiation coming from the direction of the sun and received by a surface perpendicular to that direction. Diffuse radiation is the solar radiation at the surface (horizontal) coming from all directions except that of the sun's rays. Besides total column cloud fraction we also examine cloud fraction regarding low (<2.5 km), medium (2.5<z<6 km) and high (>6 km) level clouds. Cloud fraction for each level, as well as for the total column, is calculated using the random overlapping method where the total cloud fraction $C_{rand}$ for two layers is considered as: $C_{rand} = c_a + c_b - c_a c_b$ where $c_a$, $c_b$ are the cloud fraction in each layer (Hogan and Illingworth, 2000).

We also calculate the following metrics:

1. The radiative effect of aerosol on shortwave radiation at the surface (RE). It is the difference in net shortwave radiation at the surface (netRsds) between an aerosol simulation and the CON experiment. Thus:

$$RE = netRsds_{Aerosol} - netRsds_{Control} \tag{1}$$

2. The direct radiative effect of aerosol on shortwave radiation at the surface under clear-sky conditions (cs-DRE). This is the difference in net clear-sky shortwave radiation at the surface (netCRsds) between an aerosol simulation and the CON experiment. Thus:

$$cs - DRE = netCRsds_{Aerosol} - netCRsds_{Control} \tag{2}$$

Since the cs-DRE is calculated under clear-sky conditions it encompasses only the direct aerosol effect and not the semi-direct effect.

3. The effect of clouds on shortwave radiation at the surface (SCRE). It is the difference of the net shortwave radiation at the surface (netRsds) and the net clear-sky shortwave radiation at the surface (netCRsds) for a given experiment:

$$SCRE = netRsds - netCRsds \tag{3}$$

4. In order to assess the impact of the aerosol implementation on the radiative effect of clouds, the difference of SCRE ($\Delta$ SCRE) is calculated between an aerosol experiment and CON. Therefore:

$$\Delta SCRE = SCRE_{Aerosol} - SCRE_{Control} = RE - (cs - DRE) \tag{4}$$

When comparing the group of simulations that account only for the aerosol-radiation interactions with CON, the calculated $\Delta$SCRE accounts for the semi-direct effect of aerosols.

Regarding all the variables examined, in order to assess the impact of aerosol implementation we always compare the aerosol interacting simulation to the non-interacting control simulation CON. To assess the impact of the aerosol-radiation interactions and the impact of different aerosol parameterizations, we compare the simulation family ARI, which use the Thompson2008 scheme, to CON. Comparison of the simulation ACI to CON indicates the impact of the Thompson aerosol-cloud interacting cloud microphysics scheme which implements the indirect aerosol effect. Comparison of ARCI to CON indicates the impact of both aerosol-radiation interactions and the Thompson aerosol-cloud interacting cloud microphysics scheme. Finally the only situation when a comparison is not performed against CON is when comparing ARCI to ACI, both using the aerosol-cloud interacting Thompson cloud microphysics. This enables to assess the aerosol direct and semi-direct effect under an environment where aerosol-cloud interactions (indirect effect) are also present.

The main metrics used for evaluation are Bias (model-reference), Absolute Bias (| model-reference| ) and relative Bias ((model-reference)/reference)$* 100$. Correlation coefficients between two datasets are computed using the linear Pearson correlation coefficient. Statistical significance is calculated at the 0.05 level with the Mann-Whitney non-parametric test since many of the variables examined deviate from a normal distribution. Mean daily values are used in the above tests since the time span of the simulations is not sufficient for the use monthly or seasonal values.

In order to enable grid cell comparisons of the model output against observations we use distance weighted average remapping using the four nearest neighbor values. We always remapped the finer grid onto the coarser. Therefore, all satellite products were remapped onto the WRF 0.44º grid, whereas temperature and precipitation model output was remapped onto the E-OBS 0.44º rotated grid. Furthermore, simulated temperature has been corrected with respect to the E-OBS elevation, using a temperature lapse rate of 0.65 K/km throughout the domain.

We analyze our data over the whole European domain, which we define as the as the region that consists of the Prudence subregions (Christensen et al., 2007) thus lying between -10º and 40ºin longitude and 36º to 70º in latitude. Both land and sea points are considered. Furthermore, the analysis is conducted on a seasonal basis for all four seasons of the year, winter (DJF), spring (MAM), summer (JJA) and autumn (SON). Seasonal averages are computed using mean monthly values.

## 3 Results

### 3.1 Aerosol optical depth

The mean seasonal fields of aerosol optical depth at 550nm ($AOD_{550}$) used (or produced in the case of Thompson aerosol-cloud interacting scheme) in our experiments can be seen in Fig. 1 together with the $AOD_{550}$ field of the satellite data for comparison. The fields of both simulations using the Thompson aerosol-cloud interacting scheme are very similar thus only the $AOD_{550}$ of ARCI is presented. We mainly compare against MODIS and use the SEVIRI product as an additional test. All datasets present the same basic seasonal characteristics with larger $AOD_{550}$ values during summer and spring. Exception is the field of the ARCI simulation (Thompson) that has a persistent $AOD_{550}$ maximum over Eastern Europe throughout the year and consistently presents larger $AOD_{550}$ values (0.22-0.26 range of seasonal averages) compared to all other products. The $AOD_{550}$ spatial distribution of the satellite datasets, MODIS and SEVIRI, are quite similar with MODIS presenting slightly

larger AOD$_{550}$ over continental Europe in summer (0.24 compared to 0.22). The MACC reanalysis (0.13-0.22) and MAC-v1 (0.14-0.24) climatology have a systematically higher AOD$_{550}$ on average than MODIS (comparison only over the areas with valid satellite data) with MACC being closer to the satellite product. The fact that MACC uses AOD assimilation could explain this fact. Moreover MAC-v1 has a strong and extended local maximum over Eastern Europe in summer, not seen in either

5   satellite dataset. Finally the Tegen climatology has the lowest AOD$_{550}$ (0.11-0.18) compared to the other products.

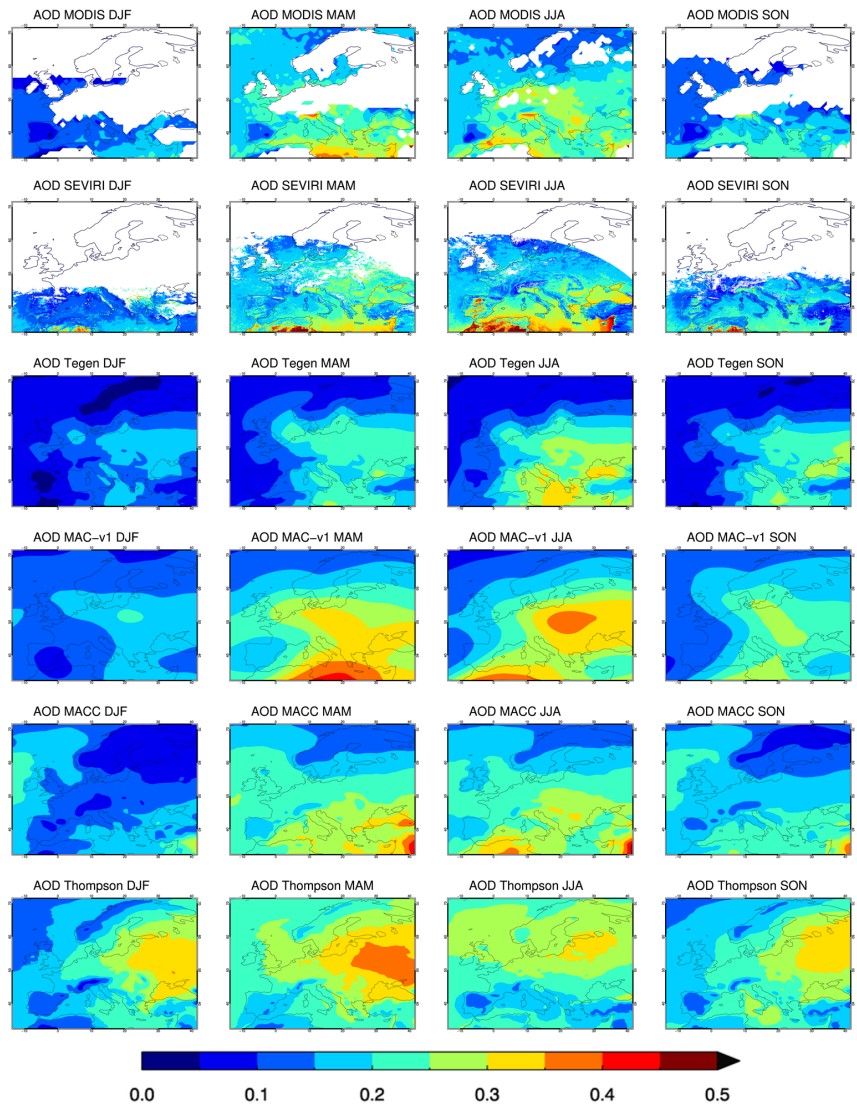

**Figure 1.** Mean seasonal aerosol optical depth at 550nm for (from top to bottom) the MODIS TERRA satellite dataset, CM SAF SEVIRI satellite dataset, the Tegen climatology, the MAC-v1 climatology, the MACC reanalysis and the ARCI simulation produced by the Thompson aerosol-cloud interacting scheme.

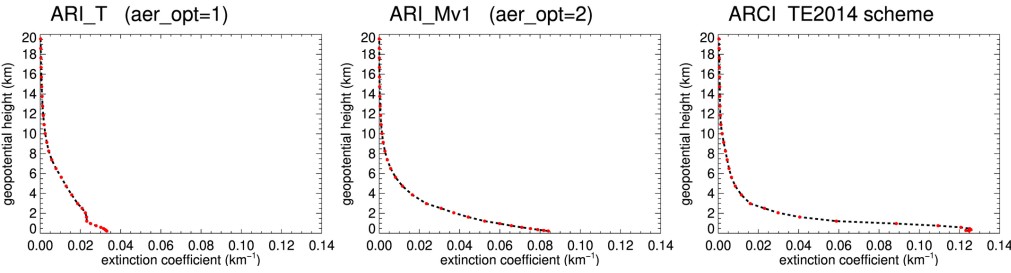

**Figure 2.** Annual mean of the domain averaged vertical distribution of aerosol extinction coefficient at 550nm (km$^{-1}$) in each model layer (red dots) for the ARI_T (indicative of aer_opt=1), ARI_Mv1 (indicative of aer_opt=2) and ARCI (indicative of the TE2014 scheme) simulations.

The vertical profile of aerosol extinction coefficient at 550nm (km$^{-1}$) (Fig. 2) has the same basic characteristics in all simulations with maximum values near the surface and a decrease of extinction coefficient with increasing altitude. The Tegen climatology in the model (aer_opt=1) has considerably less aerosol extinction near the surface than the MAC-v1 and MACC datasets used with second aerosol option (aer_opt=2), whereas the Thompson aerosol-cloud interacting microphysics scheme
5 (TE2014) has the highest near surface extinction. The Tegen climatology through the use of the first aerosol option in the model is 3-dimensional and the extinction in each model layer is calculated by the sum of extinction coefficients of each aerosol type. All the simulations using the second aerosol option (aer_opt=2) distribute the aerosol extinction vertically according to an exponential profile (Ruiz-Arias et al., 2014). Regardless of the aerosol option used, the shape of the vertical aerosol extinction profiles remains very similar for all seasons. The Thompson aerosol-cloud interacting scheme does present a somewhat larger
10 variability, but it also consistently creates a very similar profile throughout the year.

## 3.2  Evaluation of the Control Simulation

Despite some biases the control simulation (CON) captures the basic features of the European climate, which in turn indicates that the main physical processes are represented with a reasonable degree of fidelity, thus increasing the confidence on the sensitivity results.

### 15  3.2.1  Temperature

In the simulation CON winter temperatures are mostly underestimated (-0.5ºC domain average, land only), with higher cold biases over Scandinavia (despite a warm bias at the north), the Mediterranean and the Alps (-1ºC) as indicated in the upper panel of (Fig. 3). Winter cold biases especially over northern Europe are common in many EURO-CORDEX simulations (Kotlarski et al., 2014). In this study winter biases are reduced in comparison to previous WRF exercises in EURO-CORDEX
20 hindcast experiments (Katragkou et al., 2015). Since many of these WRF studies implement the Noah land surface model (Niu et al., 2011), we contend that the use of the CLM land surface model in this study is a factor for the reduced cold bias. In particular northern Europe is largely covered with snow during winter and the treatment of the snowpack by the land scheme is of particular importance. Also summer features a cold bias over most of the domain (-0.5ºC domain average) with a tendency

for minor warm biases in south and Eastern Europe. This bias pattern - cold in the north and warm in the south - has been detected in other RCM simulations over Europe such as RCA4, CCLM4, HIRHAM (Kotlarski et al., 2014).

### 3.2.2 Precipitation

Winter precipitation is overestimated throughout the domain (43% domain average), with pronounced biases existing over central (+50%) and especially over Eastern Europe, locally exceeding 100% (Fig. 3). Wet biases during DJF in Eastern Europe are common in WRF simulations (Katragkou et al., 2015; García-Díez et al., 2015; Mooney et al., 2013). The current parameterization (CON) seems to amplify the commonly simulated wet bias in the eastern part of Europe during winter. In summer biases are smaller and mostly dry (-3% domain average), which is not very typical for WRF, with most subregions presenting underestimation around -20 to -30% . However, areas with high positive relative biases are seen at the southern parts of Europe, where precipitation amounts are very small during the warm months which amplifies the relative biases. The above winter/summer bias patterns are seen in both cloud microphysics schemes used, the Thompson2008 in CON and Thompson aerosol-cloud interacting scheme. An additional simulation conducted using the WDM6 (Lim and Hong, 2010) cloud microphysics (not shown) yielded very similar results regarding precipitation bias indicating that the cloud microphysics scheme is not the main cause of precipitation bias.

### 3.2.3 Cloud fraction

Cloud fraction is overestimated in winter at 0.17 (+35% ). The relative increase is more pronounced over the Iberian Peninsula (+60% ) (Fig. 3, 3rd panel). In summer, the average overestimation is lower (0.08 or 12% ) but there is a zonal pattern with $\sim$ 30% overestimation in northern Europe and a 10 % underestimation in the Mediterranean region. However, relative biases have to be interpreted with caution in southern Europe during summertime because of the small cloud fraction amount. For both seasons similar spatial patterns, including the bias magnitudes, have been observed in other WRF simulations (Katragkou et al., 2015; García-Díez et al., 2015). In the study of Katragkou et al. (2015), the WRF simulations that had a higher cloud fraction overestimation over the northern part of the domain were the ones implementing the Grell-Devenyi cumulus parameterization. The Grell-Freitas scheme used in this study is similar to the Grell-Devenyi scheme, consequently cloud overestimation in our case could be to some extent linked to the cumulus parameterization selection, especially during summer.

### 3.2.4 Shortwave radiation to the surface and direct normalized irradiance

Shortwave downwelling radiation at the surface (Rsds) averaged for the entire European domain is underestimated for both winter and summer. In winter Rsds is in general slightly underestimated (-4% average), with some subdomains like Mid-Europe, France and British Isles reaching -20 to -40% (Fig.3). In summer the domain averaged Rsds underestimation is approximately -8%. Larger negative biases are seen in the north and decrease in intensity as we move to the south following quite closely the cloud fraction bias pattern. The cloud and Rsds bias patterns are spatially correlated, as expected. The bias pattern of direct normalized irradiance (DNI) is similar to that of Rsds but intensified. The underestimation in winter is around 13% whereas

for summer the dual pattern of underestimation to the north (-20%) and overestimation to the south (20-30%) is even more pronounced.

### 3.2.5   Evaluation of the sensitivity simulations

In general, the aerosol-interacting simulations, implementing aerosol-radiation and/or aerosol-cloud interactions and the Thompson aerosol-cloud interacting cloud microphysics, present a similar behavior to the control simulation CON, regarding the biases of the main variables described above. This indicates that aerosol representation, despite its considerable impact seen in the next chapter, is not the main source of bias in our simulations. Moreover, aerosol introduction, despite making the representation of physical processes in the model more complete, often does not lead to bias improvements. Furthermore the improvement of bias does not necessarily mean that the aerosol representation is correct, since model biases can be the result of compensation between errors in the aerosol representation and errors induced by other physical mechanisms (García-Díez et al., 2015). Zubler et al. (2011) in an RCM study reached similar conclusions, stating that the overestimation of aerosol optical depth was responsible for masking strong biases in the simulated cloud fraction. Figure S2 in the supplement presents the basic biases for simulation ARI_T with the Tegen climatology.

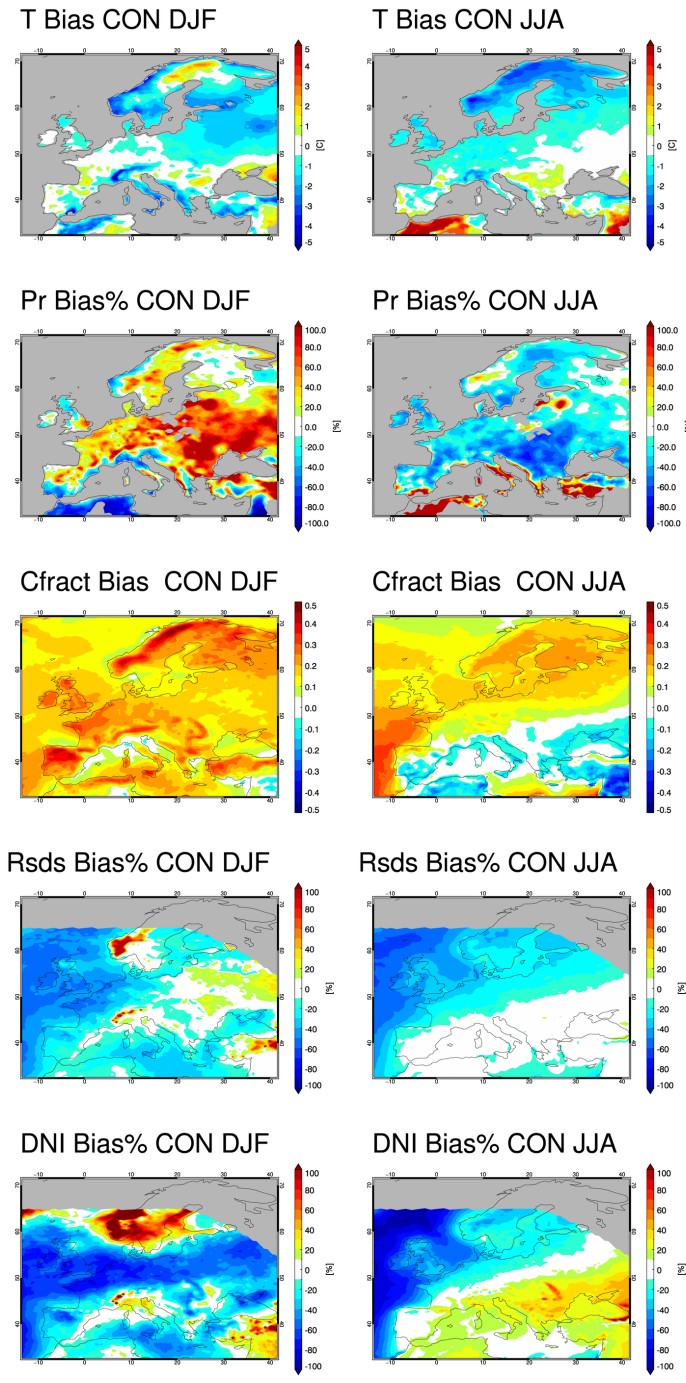

**Figure 3.** Bias plots for control simulation CON for winter (DJF-left) and summer (JJA-right). Biases depicted from top to bottom for temperature (T), precipitation (Pr), total cloud fraction (Cfract), downwelling shortwave radiation to the surface (Rsds) and direct normalized irradiance at the surface (DNI).

### 3.3 Aerosol-radiation interactions

In this section we explore the impact of only aerosol-radiation interactions implementation in the model. Thus we present results for the ARI group simulations.

#### 3.3.1 Clear sky radiation at the surface

Accounting for the aerosol radiation interactions leads to statistically significant reductions in clear sky downwelling shortwave radiation to the surface (Crsds). Crsds decreases by 5 to 8% (domain average), depending on the simulation, during all seasons. Larger reductions of 14% are found in the ARI_Mv1urban simulation. Figure 4 shows the clear-sky direct radiative effect (cs-DRE) at the surface quantified as the difference of netCrsds between each simulation and the control (CON). The domain averaged cs-DRE when aerosol-radiation interactions are enabled is very similar, despite the different aerosol datasets, for

all ARI simulations and is around -4 to -5W/m$^2$ in winter and -14 to -17W/m$^2$ in summer (Table 2). ARI_Mv1urban shows twice the reduction than other aerosol treatments due to the considerably more absorbing nature of "urban" type aerosols. Spatially the cs-DRE correlates very well with the AOD$_{550}$ field of each simulation, with the AOD$_{550}$ maxima coinciding with the Crsds minima for each experiment. Spatial correlation coefficients for the ARI group range between -0.8 and -0.98. The Tegen climatology used in ARI_T leads to a similar clear-sky shortwave radiation decrease with the rest ARI group simulations

(except ARI_Mv1urban) despite the fact that the AOD$_{550}$ of Tegen is considerably smaller than that of MAC-v1 or MACC. It must be noted however that the ARI_T simulation has lower single scattering albedo (SSA) values and thus more absorbing aerosol than all the ARI group simulations, except ARI_Mv1urban. Because of the lower SSA, the ARI_T simulation produces a larger decrease of clear-sky radiation per unit of AOD$_{550}$ (W/m$^2$/AOD) and thus despite the smaller AOD$_{550}$ it presents a similar direct radiative effect.

#### 3.3.2 Radiation at the surface

Shortwave downwelling radiation at the surface (Rsds) shows significant attenuation almost all over the domain throughout the year. Domain averaged Rsds reduction lies in the range -3 to -8% for all seasons, quite similar with the decrease seen in clear-sky radiation (Crsds). ARI_Mv1urban is again an exception with higher attenuation around -12 to -16%.

The change in the net shortwave radiation at the surface constitutes the radiative effect (RE) of aerosol (Fig. 5) and comprises

of the clear-sky direct radiative effect (cs-DRE) and the effect on radiation due to changes in cloud amount and properties ($\Delta$SCRE). Accounting for aerosol-radiation interactions only, leads to a negative RE of -2W/m$^2$ in winter and -11 to -13 W/m$^2$ in summer (-7W/m$^2$ annual average) with ARI_Mv1urban roughly doubling these values (Table 2). Compared to other studies, our results present in general a smaller radiative effect over Europe. Nabat et al. (2015) showed an annual average RE of -10 W/m$^2$. The study of Huszar et al. (2012) calculated a similar to our study RE during summer (-12 to -15 W/m$^2$)

but a considerably larger effect (-7 W/m$^2$) in winter whereas the RegCM3 study of Zanis (2009) for the year 2000 presented a higher summer radiative effect (-16 W/m$^2$). When implementing only aerosol-radiation interactions, the spatial correlation between the radiative effect RE and the AOD$_{550}$ field is high (-0.6 to -0.9).

It is important to note that aerosol optical properties besides AOD can have a severe impact on seasonal radiation amounts. For example, simulations ARI_Mv1, ARI_Mv1full and ARI_Mv1urban all use the MAC-v1 AOD550 data but parameterize the other aerosol optical properties differently. ARI_Mv1 and ARI_Mv1full have similar single scattering albedo (SSA) values at the visible spectrum (0.92 to 0.98) which leads to similar results in domain averaged Rsds decrease. ARI_Mv1urban however has considerably more absorbing aerosols (SSA starting from 0.6) leading to an almost doubled impact on Rsds attenuation. This impact is widespread over the domain with the overall distribution of Rsds decrease being clearly shifted towards more negative values (Fig. S3). Alexandri et al. (2015) also stressed the importance of secondary aerosol parameters such as SSA in simulating solar radiation in regional climate simulations.

We have seen that the impact of aerosol-radiation interactions is important in shortwave radiation at the surface. However it is even more pronounced in its direct and diffuse components. Direct normalized irradiance (DNI) is reduced much more severely than Rsds in the ARI group of experiments. Since DNI comes only from the direction of the sun, any interaction with aerosol (scattering, absorption) removes radiation amounts from this direction. On the other hand, Rsds is reduced only when it is absorbed or scattered at an angle that does not reach the surface. Thus the aerosol direct effect is much stronger in DNI. It is characteristic that compared to control CON, domain averaged differences are around -30% for all seasons. Locally attenuation can even exceed -50%, especially during winter and autumn where DNI levels are low due to large cloud amounts and small overall radiation levels.

Contrary to DNI, diffuse radiation is strongly increased with aerosol-radiation interactions. Diffuse radiation reaches the surface from all angles except from the direction of the sun (direct radiation). Thus when direct radiation is scattered by aerosol, a part of it transforms into diffuse radiation and therefore increases the diffuse radiation component. As expected, this effect causes an increase in diffuse radiation in almost all simulations, the exception being the ARI_Mv1urban simulation which has a large decrease in cloud fraction (see Figure 6). The amount of DIF relative increase varies considerably with seasons. For winter it is around 7 to 20% and for summer it is around 30 to 40%. The impact of aerosols in DIF is generally more pronounced over areas with low cloud amounts such as southern Europe during summer. The much stronger impact on DNI and DIF makes it essential to examine these variables in conjunction with Rsds, in order to fully understand the impact of aerosol-radiation interactions on radiation.

### 3.3.3 Total cloud fraction and cloud radiative effect

Changes in total cloud fraction (CFRACT) compared to CON due to aerosol implementation are shown in Fig. 6. In general, regardless of the type of aerosol implementation changes are quite small. Therefore, domain averaged differences from CON do not exceed 0.01, (scale of 0 to 1). This partially happens because cloudiness increases and decreases in parts of the domain. However, the averaged absolute differences from CON are still quite small with a range of 0.01 to 0.03. The smallest impact is seen in winter where cloudiness is mainly affected by synoptic phenomena. In relative values, domain changes are around 1-2% for winter and up to 3-4% (6% for ARI_Mv1urban) during summer. In ARI_Mv1urban CFRACT changes exceed in some cases 0.15. The aerosol-radiation interaction has a minor impact on CFRACT. Some areas show statistically significant differences in CFRACT which follow the pattern of temperature changes. In several cases, cloud fraction increase occurs in areas with strong

near surface temperature decrease (e.g. north of the Black Sea in autumn and over central Europe during summer in ARCI-ACI) whereas decreases in cloud cover are related to areas with strong atmospheric warming (e.g. ARI_Mv1urban over the Alps in summer). The most pronounced CFRACT increases occur above the Black Sea and eastern Balkans in autumn (including parts of North Africa and Central-Eastern Mediterranean in some cases). These changes are present in all the simulations (Fig. 6). They are probably related to the formation of a cyclonic anomaly in the wind field (both 850 and 500hPa) over the Black Sea region (Fig. S4). The introduction of aerosol-radiation interactions reduces radiation at the surface, thus decreasing temperature. Close to the maximum of cooling a cyclonic anomaly is formed and larger cloud fraction amounts are produced which in turn further decreases radiation levels hence decreasing temperature, indicating a possible feedback mechanisms (Fig. S5). Extended parts of this cyclonic anomaly are of statistical significance mainly in simulations ARI_T and ARI_Mv1urban. However, this is not the case for all the ARI simulations. Also the intensity of the cyclonic anomaly varies considerably between simulations. Therefore, the internal model variability as well as the real climate variability could be very important in this kind of complex feedback mechanisms. The use of different physics parameterizations, initial conditions and even different time periods may have a large impact and could potentially modify this cyclonic anomaly effect. The influence of aerosols on the South Asian monsoon is well recognised (Bollasina et al., 2014; Ganguly et al., 2012) and it would be interesting to explore whether this cyclonic anomaly effect might also be an aerosol-circulation effect important for European weather and climate. The impact on cloudiness is more pronounced in ARI_Mv1urban as a result of extreme absorbing aerosols. In this simulation, significant changes in CFRACT are found in extended parts of the domain for all seasons except winter. This highlights the importance of introducing aerosol optical properties (e.g. SSA) in RCM simulations, as they can affect the thermodynamics of the lower and mid troposphere (Fig. S6). The patterns of significant changes in total cloud fraction in our simulations are dominated by changes in low clouds, which are most affected. Medium level cloud changes are less pronounced in amplitude and area extent, whereas higher clouds are least impacted by changes in aerosol treatments. This is to be expected, since the specified aerosol concentrations are located in the lower part of the troposphere.

We showed that accounting for the aerosol-radiation interactions does not systematically change CFRACT. Of particular interest is the impact of aerosol on the ability of clouds to interact with radiation. To study this effect we calculate the aerosol-related change in the cloud radiative effect regarding shortwave radiation at the surface ($\Delta$SCRE) (Fig. 7). The domain averaged change in the cloud effect on radiation is positive in all experiments (Table 2). Thus, the introduction of aerosol-radiation and/or aerosol-cloud interactions leads to cloudiness enabling larger amounts of radiation to reach the surface. This can happen due to changes in cloudiness amount or in cloud optical properties. Since there is no general decrease of cloud fraction amount in the ARI simulations (except in ARI_Mv1urban) the positive $\Delta$SCRE must be attributed to changes in the optical properties of clouds. For the ARI simulations, $\Delta$SCRE represents the impact of semi-direct aerosol effect on radiation, which is positive with annuals averages around 3 to 4 W/m$^2$ is largest during spring (5-7 W/m$^2$). Nabat et al. (2015) had calculated a larger annually averaged semi-direct effect around 5 to 6 W/m$^2$. This effect is counteracting the clear-sky direct radiative effect (cs-DRE) of aerosol that is clearly negative. The semi-direct effect accounts for 60% of the direct aerosol effect on radiation (cs-DRE) during winter, 45% during spring and around 20-35% during summer and autumn. Consequently, the impact of

semi-direct effect on radiation is considerable, and plays an important role in the overall impact of aerosol-radiation interaction implementation in the model.

**Table 2.** Domain averages for each season regarding aerosol optical depth at 550nm (AOD$_{550}$), Radiative effect (RE), clear-sky direct radiative effect (cs-DRE) and change in shortwave cloud effect at the surface (ΔSCRE) all calculated as differences from control CON. For all experiments. At the first column the aerosol effect that is being implemented is stated above each group of simulations. For simulation ARCI all the above quantities are also calculated against ACI (e.g. ARCI-ACI) in order to assess the implementation of aerosol-radiation interactions in the Thompson aerosol-cloud interacting cloud microphysics.

| Radiation interacting | AOD | | | | RE | | | | cs-DRE | | | | ΔSCRE | | | |
|---|---|---|---|---|---|---|---|---|---|---|---|---|---|---|---|---|
| | DJF | MAM | JJA | SON | DJF | MAM | JJA | SON | DJF | MAM | JJA | SON | DJF | MAM | JJA | SON |
| ARI_T | 0,11 | 0,16 | 0,18 | 0,15 | -2 | -7 | -13 | -7 | -5 | -13 | -16 | -9 | 3 | 7 | 4 | 2 |
| ARI_Mv1 | 0,14 | 0,24 | 0,24 | 0,19 | -2 | -8 | -12 | -5 | -4 | -13 | -15 | -8 | 3 | 5 | 4 | 3 |
| ARI_Mv1urban | 0,14 | 0,24 | 0,24 | 0,19 | -4 | -18 | -26 | -12 | -8 | -29 | -34 | -16 | 4 | 11 | 8 | 4 |
| ARI_Mv1full | 0,14 | 0,24 | 0,24 | 0,19 | -2 | -8 | -13 | -5 | -5 | -14 | -17 | -9 | 3 | 6 | 4 | 4 |
| ARI_MC | 0,13 | 0,22 | 0,22 | 0,17 | -2 | -6 | -11 | -5 | -4 | -12 | -14 | -7 | 2 | 6 | 3 | 2 |
| ARCI-ACI | 0,22 | 0,26 | 0,24 | 0,23 | -1 | -6 | -11 | -3 | -5 | -13 | -14 | -8 | 4 | 7 | 4 | 5 |
| **Cloud interacting + cloud microphysics** | | | | | | | | | | | | | | | | |
| ACI | - | - | - | - | 2 | 7 | 10 | 3 | 0 | 0 | 0 | 0 | 2 | 6 | 10 | 3 |
| **Radiation + Cloud interacting + cloud microphysics** | | | | | | | | | | | | | | | | |
| ARCI | 0,22 | 0,26 | 0,24 | 0,23 | 1 | 0 | -1 | 0 | -5 | -13 | -14 | -8 | 6 | 13 | 13 | 8 |

### 3.3.4 Temperature

Accounting for the aerosol-radiation interactions (ARI group) leads to surface cooling, as expected due to the lower radiation levels reaching the ground. Domain averaged changes compared to CON are negative and range between -0.1 to -0.3 °C (annual averages) with the largest impact seen during summer and autumn (Table 3). These values are very similar to those in the RegCM study over Europe of Zanis et al. (2012). If we calculate the change only over land, then temperature is further decreased and ranges between -0.2 to -0.4 °C (annual averages). The lack of coupling with an ocean model limits the effect of temperature change over sea in our simulations. The study of Nabat et al. (2015) presents a cooling of -0.4 °C (annual average) over land. Finally the temperature impact in localized areas can be considerably higher, in cases reaching a decrease of 1.5 °C. Cases of such strong reduction are limited in spatial extent and are seen mainly in summer and autumn within the areas of intense cooling like the Balkans and near of the Black Sea. Despite the larger AOD$_{550}$ in summer, the temperature impact is greater in autumn. This is probably related to the fact that the relative Rsds decrease is slightly larger in autumn (except for ARI_Mv1full). It is also interesting to note that differences in the single scattering albedo can have an effect on temperature at the surface despite the use of the same AOD$_{550}$ field. This is the case not only when changing considerably

the SSA values (e.g. ARI_Mv1urban) but also when more moderate changes are implemented. For example ARI_Mv1 and ARI_Mv1full have SSA values within a very similar range, however ARI_Mv1full presents larger temperature decrease (-0.4 °C) compared to ARI_Mv1 (-0.2 °C). The temperature decrease is not constrained to the surface but is also detected at higher levels, with decreasing intensity at higher altitudes, usually reaching 850hPa. In the case of autumn over the Balkans and the Black Sea a decrease of -0.2 °C can be seen almost up to 400hPa (Fig. S5). In summer, ARI_Mv1urban is the only simulation from the ARI group that presents a large area of statistically significant warming at the surface, seen over parts of the Alps, the Iberian Peninsula, Italy and the Balkans, coinciding with a decrease in total cloud fraction (CFRACT). This warming can be attributed to the highly absorbing "urban" type aerosols that warm the atmosphere by absorbing solar radiation but also can affect temperature through circulation and cloud cover amount changes (Fig. S6). This temperature increase clearly affects the surface but also reaches higher levels up to 200hPa. The aerosol absorptivity, expressed through the SSA, can have a strong effect on the signal of the temperature changes presented. Warming of near surface temperature, including the pattern described above during summer (with slightly smaller warming), has also been described by other studies (Huszar et al., 2012; Zanis, 2009) that implemented much more realistic and less absorbing aerosols compared to ARI_Mv1urban. We must remind here that ARI_Mv1urban is more of an idealized experiment with unrealistically absorbing aerosol.

**Table 3.** Domain averaged temperature difference (°C) compared to CON for all experiments and seasons. In parenthesis the values when only land points are considered. Where stated, for simulation ARCI the above quantities are also calculated against ACI (ARCI-ACI ) in order to assess the implementation of direct effect in the Thompson aerosol-cloud interacting cloud microphysics.

| (°C) | Year | | DJF | | MAM | | JJA | | SON | |
|---|---|---|---|---|---|---|---|---|---|---|
| ARI_T | -0,2 | (-0,3) | -0,1 | (-0,1) | -0,1 | (-0,2) | -0,2 | (-0,4) | -0,4 | (-0,5) |
| ARI_Mv1 | -0,2 | (-0,2) | -0,1 | (-0,1) | -0,1 | (-0,2) | -0,3 | (-0,4) | -0,2 | (-0,3) |
| ARI_Mv1urban | -0,2 | (-0,4) | -0,2 | (-0,3) | -0,1 | (-0,2) | -0,2 | (-0,4) | -0,4 | (-0,6) |
| ARI_Mv1full | -0,3 | (-0,4) | -0,1 | (-0,2) | -0,3 | (-0,4) | -0,3 | (-0,5) | -0,3 | (-0,4) |
| ARI_MC | -0,1 | (-0,2) | -0,1 | (-0,1) | 0,0 | (-0,1) | -0,1 | (-0,2) | -0,2 | (-0,3) |
| ARCI-ACI | -0,1 | (-0,2) | -0,2 | (-0,3) | -0,1 | (-0,2) | -0,2 | (-0,3) | 0,1 | (0,1) |
| | | | | | | | | | | |
| ACI | 0,1 | (0,1) | 0,1 | (0,1) | 0,1 | (0,2) | 0,2 | (0,3) | -0,1 | (-0,1) |
| ARCI | 0,0 | (0,0) | -0,1 | (-0,2) | 0,0 | (0,0) | 0,0 | (0,0) | 0,0 | (0,0) |

### 3.3.5 Precipitation

Aerosol related domain averaged changes of precipitation are small in most experiments (±0.08mm/day), at most up to ±5% in relative values. ARI_Mv1urban again has a more intense impact with a relative decrease of around -13% (-0.2m/day) in JJA and MAM. All the other ARI experiments have no specific tendency of precipitation change throughout the year. However in spring and summer most of the ARI group simulations (except ARI_Mv1full) have a small domain averaged precipitation decrease (-2 to -5%, -0.02 to -0.09 mm/day). In general winter is the season which is least impacted by aerosol implementations. The

study of (Nabat et al., 2015) using a coupled atmospheric-ocean model showed a decrease in precipitation over Europe. This decrease was attributed to the aerosol induced cooling of sea surface temperature (SST) that led to decrease latent heat fluxes consequently decreasing atmospheric humidity and cloud cover. Therefore the use of prescribed SST in the current study can be seen as a limitation and could particularly affect precipitation results. The small domain averages are to an extent a product

of sign compensation since the spatial pattern of precipitation differences from control is not homogenous but consists of small areas with increases and decreases scattered around the domain. Precipitation changes at a grid scale level in some cases can exceed $\pm 50\%$. However, this effect can probably be attributed to internal model variability and not to aerosol implementation. A common area of significant precipitation increase in all experiments is seen over the Black Sea in autumn, where a significant CFRACT increase and cyclonic anomaly in the wind field at 850 and 500hPa is present. This characteristic cyclonic anomaly

(Fig. S4) is seen in all ARI group simulations but also to a lesser extent in simulations ACI and ARCI (not shown). There is no clear spatial correlation between changes in cloud amount and changes in precipitation. Over the Black Sea in autumn increase in precipitation coincided with increase in CFRACT. It should be reminded however, that the simulations do not have an ocean-atmosphere coupling, something that can influence the results on precipitation over the Black Sea. ARI_Mv1urban exhibits the largest and the spatially most extensive impact on precipitation. During summer and spring large areas of precipitation

decrease are seen over Central-Southern Europe and the Balkans coinciding spatially with CFRACT decrease (see section 3.3.3). Clearly, the warming of the mid troposphere due to the highly absorbing nature of the aerosols in ARI_Mv1urban stabilizes the atmosphere leading to both precipitation suppression and cloud dissolution.

## 3.4   Aerosol-radiation interactions with aerosol-cloud interactions present

In this section we examine the impact of aerosol-radiation interactions when the aerosol-cloud interactions are also present.

For this purpose we compare simulation ARCI that has aerosol-radiation and aerosol-cloud interactions, to simulation ACI that has only aerosol-cloud interactions. Both simulations use the Thompson aerosol-cloud interacting cloud microphysics (Thompson and Eidhammer, 2014). In general, the behaviour of aerosol-radiation interactions in general circulation model simulations where the aerosol-cloud interaction effects are also represented is quite similar to the implementation of only aerosol-radiation interactions. The main difference is that the change in the cloud radiative effect ($\Delta$SCRE)becomes even more

positive, compared to the ARI group of simulations. Therefore clouds let even more radiation to reach the surface and thus further reduce the direct effect of aerosol. In this case $\Delta$SCRE (4-7 W/m$^2$) is slightly larger (1-3 W/m$^2$) compared to the effect in the ARI group (Table 2), and its relative importance is also increased, amounting up to 80% of the direct effect of aerosol cs-DRE in winter and 65% in autumn. Interestingly the positive changes in the cloud radiative effect are more pronounced over the Atlantic Ocean at the north-west part of the domain during summer (Fig. 7). The more positive $\Delta$SCRE leads to a smaller

reduction of shortwave radiation at the surface and a less negative aerosol radiative effect RE (-1W/m$^2$ in DJF, -11W/m$^2$ in JJA). The components of shortwave radiation are also impacted. Direct normalized irradiance is reduced but to a lesser extend (-20% in all seasons) compared to the implementation of aerosol-radiation interactions only (ARI group). Diffuse radiation increases (6 to 26%) during all seasons but this increase is also smaller than the ARI group. The positive changes in cloud radiative effect are again not driven by changes in cloudiness since there is no overall cloud fraction reduction. On the contrary, in summer

over central Europe there is a statistically significant cloud fraction increase. However, cloud fraction changes between ARCI and ACI are generally small and do not exceed the changes seen when implementing only aerosol-radiation interactions. As expected, the overall decrease in shortwave radiation at the surface leads to a decrease in near surface temperature. However, the smaller radiation reduction at the surface, compared to the ARI group, does not particularly influence this temperature decrease.

For most seasons, the cooling is very similar to the one seen when only the aerosol-radiation interactions are implemented. An exception is autumn where a weakened aerosol radiative effect (RE) seems unable to produce a clear temperature decrease over the domain. Regarding precipitation, in contrast to the ARI group that exhibited no specific behavior, domain averaged precipitation is slightly reduced for all seasons except spring. This is more pronounced in autumn. However, the spatial pattern of precipitation changes is still quite noisy and does not present a specific behavior over the entire domain.

**3.5   The Thompson aerosol aware-scheme**

In this section we explore the impact of the Thompson aerosol-cloud interacting cloud microphysics scheme compared to the Thompson2008 scheme that has no aerosol-cloud interactions. The choice of microphysics scheme has an impact on cloudiness. The two simulations using the aerosol-cloud interacting cloud microphysics (ARCI and ACI) have lower cloud fraction amounts throughout the year compared to control CON and all other simulations using the Thompson2008 scheme.

This is probably connected to the fact that the above two simulations also present smaller liquid water path (LWP) values. The smaller cloud fraction amount has an impact in the cloud effect on radiation. Of course the changes in the cloud radiative effect compared to control simulation CON are not only attributed to the change in the microphysics scheme. In the case of ACI they are also attributed to the enabled aerosol-cloud interactions and in ARCI to both aerosol-cloud and aerosol-radiation interactions. Simulation ACI has a positive change in cloud radiative effect at the surface ($\Delta$SCRE) compared to

CON throughout the year. Therefore if we compare ACI, that has no aerosol-radiation interactions, to control simulation (CON) we see that ACI presents an increase of shortwave radiation at the surface and thus a positive RE (2 to 10 W/m$^2$ depending on season). This results in a domain averaged temperature increase (0.1 to 0.2$^o$C) compared to CON for all seasons except autumn. In simulation ARCI the use of aerosol-radiation interactions further increases the positive change in the cloud radiative effect (as we have seen in the ARCI-ACI comparison). Thus, ARCI presents by far the largest increase in cloud

radiative effect against control between all the simulations of this study. Therefore, if we compare ARCI to CON we observe that ARCI presents a close to zero radiative effect (RE) throughout the year. Clear sky radiation is decreased and cs-DRE (-5 to -14 W/m$^2$) is negative due to the aerosol-radiation interactions. However, the large positive change in cloud radiative effect (6 to 13 W/m$^2$) ($\Delta$SCRE) compensates for the decrease in clear-sky radiation and leads to negligible changes in the domain averaged overall shortwave radiative effect. Spatially the RE includes both positive and negative values, with the positive ones being more intense in the northern and western part of the domain during summer and spring. Regarding the indirect

aerosol effect, the study of Da Silva et al. (2018) used the Thompson aerosol-cloud interacting cloud microphysics scheme to experiment with different aerosol concentrations and showed that increased aerosol loads decreased summer precipitation amounts. Our study did not experiment with different aerosol loads and thus it does not make statements regarding solely the impact of the aerosol indirect effect. Finally, it must be noted that the implementation of the Thompson aerosol-aware scheme

in the model resulted in a minimal computational cost increase (+10%) compared to the Thompson2008 scheme. Therefore, the aerosol-aware scheme presents a very fast option to incorporate interactive aerosol in WRF with aerosol-radiation and aerosol-cloud interaction capabilities.

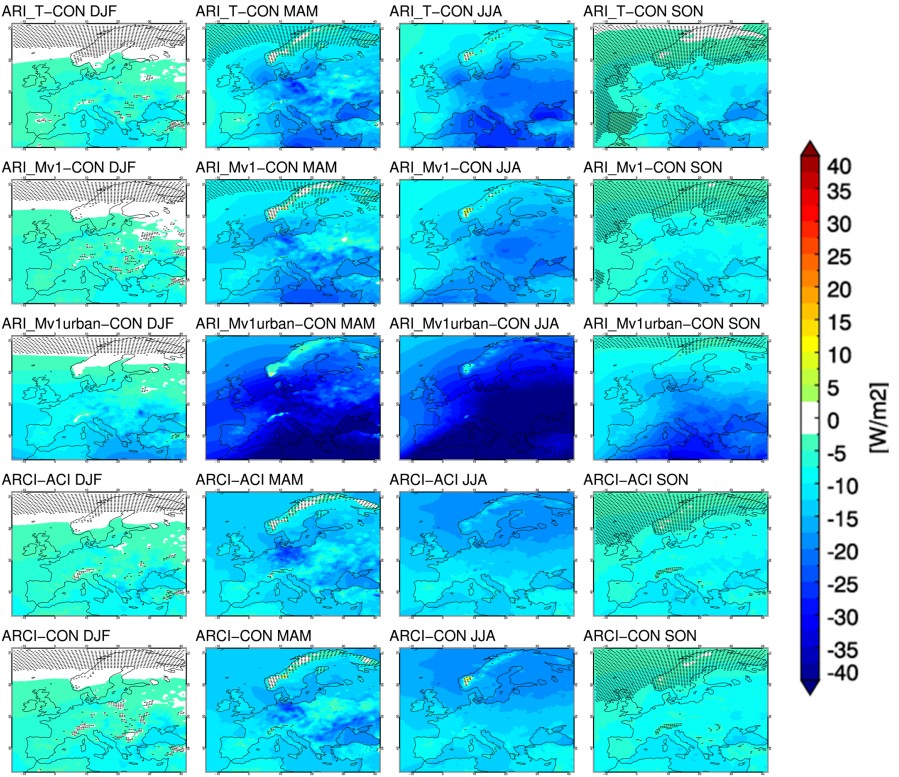

**Figure 4.** Clear-sky direct radiative effect (cs-DRE) at the surface for simulations implementing aerosol-radiation interactions for all seasons. cs-DRE has been calculated as the difference in net Crsds at the surface from control CON for the ARI group of simulations (rows 1 to 5). The last row depicts the aerosol-radiation interactions in an environment where the indirect effect is also present and displays the difference of experiment ARCI from ACI. Stippling indicates areas where the differences are not statistically significant at the 95% level, according to the Mann-Whitney non-parametric test.

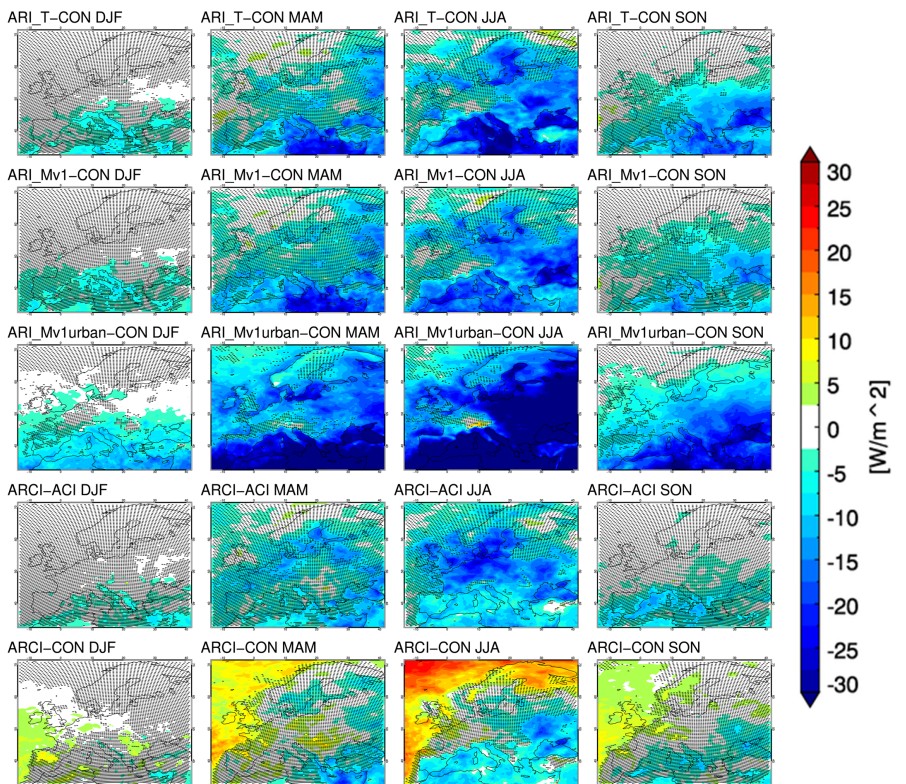

**Figure 5.** Radiative effect (RE) calculated against control CON for all experiments and seasons. Furthermore, the RE of ARCI calculated against ACI (ARCI-ACI) is given to assess aerosol-radiation interaction implementation in the Thompson aerosol-cloud interacting cloud microphysics (TE2014) (row six). First six rows present the impact of direct effect. Last two rows (black box) present the impact of TE2014 with indirect effect against control (row seven) and TE2014 with aerosol-radiation interaction enabled against control (row eight). Stippling indicates areas where the differences are not statistically significant at the 95% level, according to the Mann-Whitney non-parametric test.

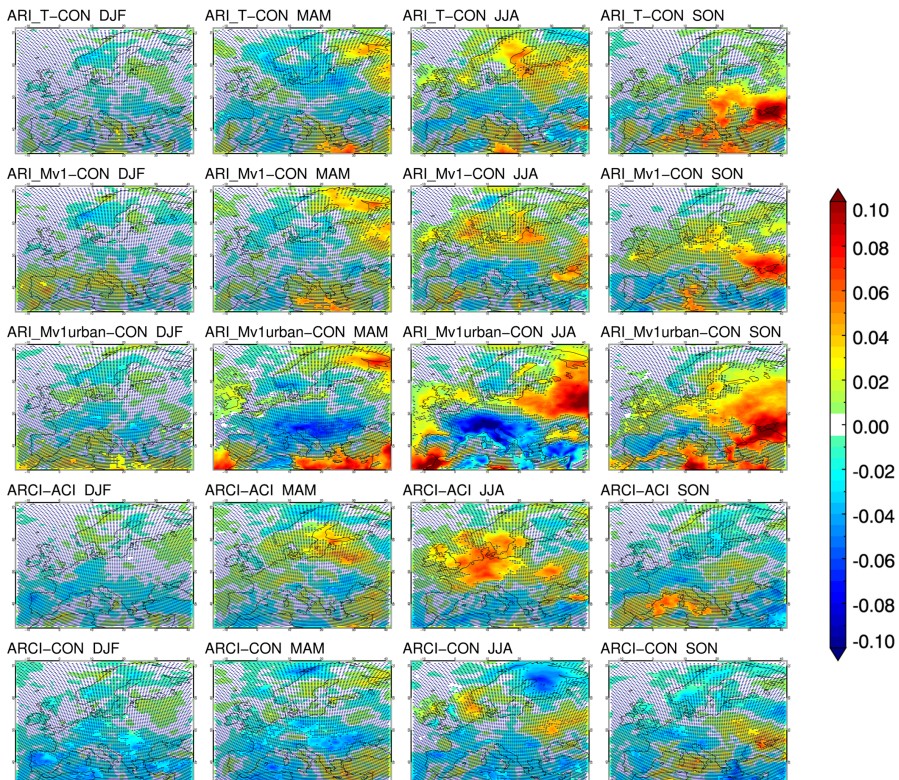

**Figure 6.** Total cloud fraction (CFRACT) difference from control simulation CON for all experiments and seasons. Furthermore the CFRACT difference of ARCI calculated against ACI (ARCI-ACI) is given to assess aerosol-radiation interaction implementation in the Thompson aerosol-cloud interacting cloud microphysics (TE2014) (row six). First six rows present the impact of aerosol-radiation interactions. Last two rows (black box) present the impact of TE2014 with indirect effect against control (row seven) and TE2014 with aerosol-radiation interactions enabled against control (row eight). Stippling indicates areas where the differences are not statistically significant at the 95% level, according to the Mann-Whitney non-parametric test.

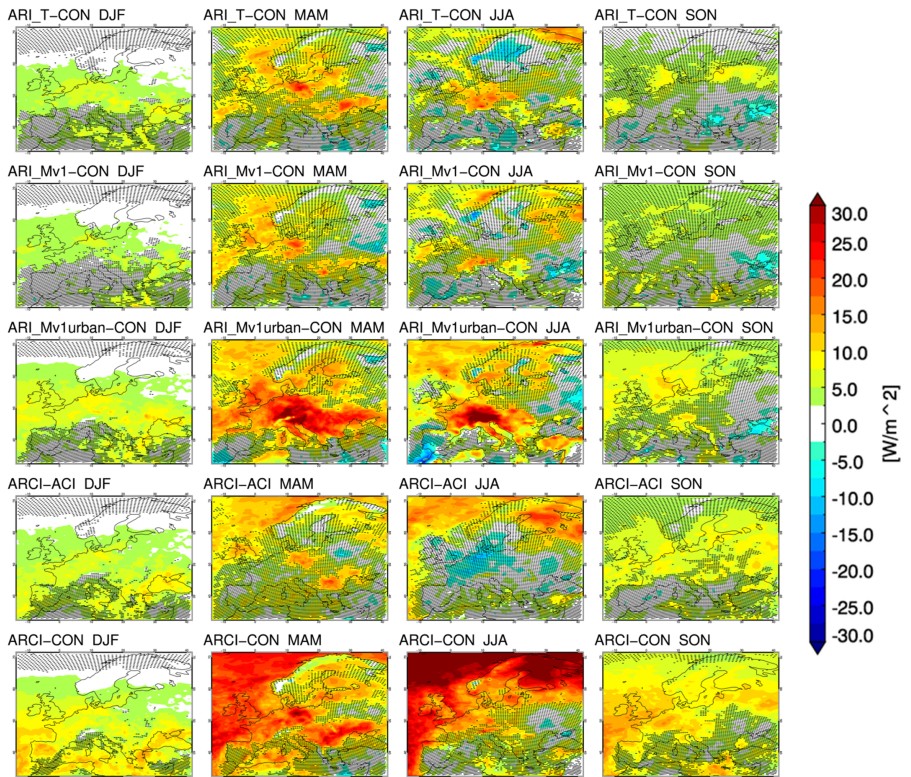

**Figure 7.** Shortwave cloud radiative effect difference (ΔSCRE) from control simulation CON for all experiments and seasons. Furthermore the SCRE difference of ARCI calculated against ACI (ARCI-ACI) is given to assess aerosol-radiation interaction implementation in the Thompson aerosol-cloud interacting cloud microphysics (TE2014) (row six).The last two rows (black box) present the impact of TE2014 with indirect effect against control (row seven) and TE2014 with aerosol-radiation interactions enabled against control (row eight). Stippling indicates areas where the differences are not statistically significant at the 95% level, according to the Mann-Whitney non-parametric test.

## 4   Conclusions

In this study, we have explored the sensitivity of resolving aerosol interactions within downscaling regional climate model experiments over Europe using different aerosol products and different modeling options to couple aerosol with model physics accounting mainly for the aerosol-radiation interactions but also including aerosol-cloud interactions in two simulations. The aerosol input we tested included older climatologies widely used in climate studies (e.g. Tegen, 1997) and relatively newer products (e.g. ECMWF MACC reanalysis), which have not been extensively tested yet by the RCM community. These new datasets are promising due to their higher spatial and temporal resolution. The different experiments and configurations applied in our model simulations allow for i) the quantification of the direct and semi-direct aerosol effect over Europe and ii) the assessment of the impact of aerosol parameterization (AOD, ASY, SSA) and type (absorbing vs non-absorbing) on regional

climate. Our model results show that the aerosol-radiation interactions in the model have a clear and significant impact (-3 to -16%) on shortwave radiation at the surface (Rsds) throughout the year, whereas the influence on direct normalized irradiance (-30%) and diffuse radiation (+10 to +40%) can be considerably stronger. These findings are particularly important for solar applications (e.g., solar power production), since Rsds is often the only available parameter from ensemble climate projects

(e.g., CORDEX; e.g. Jerez et al.,2015), although it is neither the most sensitive to aerosol properties nor the most relevant for the impact community (Jimenez et al., 2016). Accounting for the aerosol-radiation interactions reduces surface radiation by up to -17 (-5) W/m$^2$ in summer (winter) due to the clear-sky direct radiative effect (cs-DRE). This reduction is twice as large for aerosol of highly absorbing nature (here in the simulation with the "urban" aerosol type). In all simulations enabling aerosol-radiation interactions, clouds responded (semi-direct effect) by letting more radiation to reach the surface (positive

change in cloud radiative effect). This effect must be attributed to changes in the optical properties of clouds since a general decrease of cloud fraction amount is not detected. This positive change in the cloud radiative effect considerably counteracts the impact of the cs-DRE by 20 to 60% (2 to 4 W/m$^2$) depending on season. Therefore, the overall radiative effect of aerosols (RE) is clearly smaller than the cs-DRE and is approximately -12 (-2) W/m$^2$ in summer (winter). Similar studies implementing aerosol-radiation interactions have calculated larger values of both overall radiative effect (Nabat et al., 2015; Huszar et al.,

2012; Zanis, 2009) and semi-direct effect. Furthermore, when aerosol-radiation interactions are implemented in a simulation where the aerosol-cloud interactions are also introduced, the combined impact of the semi-direct and indirect effects results in an even more positive change in cloud radiative effect (4 to 7 W/m$^2$), thus further weakening the overall aerosol radiative effect (–1 W/m$^2$ in winter and -11 W/m$^2$ in summer). The decrease of shortwave radiation at the surface due to aerosol-radiation interactions leads to a widespread temperature decrease with domain-averaged cooling reaching -0.5$^o$C over land in summer

and autumn. Locally the cooling can be considerably stronger, reaching -1.5$^o$C close to the maxima of aerosol optical depth. The impact on temperature decreases with height and is detectible at least up to the 850 hPa pressure level. The idealized experiment with the extremely absorbing "urban" type aerosols leads also to near surface cooling which is now accompanied by an intense tropospheric warming at higher altitudes, in cases exceeding 2$^o$C (around the 700 hPa level). We have also shown that introducing the aerosol-radiation and aerosol-cloud interactions may disturb the climate system in a way that affects

cloudiness (especially low-level cloudiness) with the potential to trigger regional circulation anomalies at the lower and the mid-troposphere. Precipitation was not particularly affected by most of the aerosol perturbations in our 5 year simulations. The spatial pattern of the changes is patchy and some large local changes are probably a result of internal model variability. However in spring and summer a small domain-averaged precipitation decrease (-2 to -5%, -0.02 to -0.09 mm/day) is seen. The study of (Nabat et al., 2015), investigating aerosol-radiation interactions, found a precipitation reduction for all seasons,

due to the decrease of SST, which in turn lead to reduced evaporation and reduced cloud fraction and precipitation. That study however used an RCM coupled with an ocean model, which made possible to simulate changes in the SST, a component that our study is missing. In our study, considerable precipitation reduction over extended areas is seen only with the use of highly absorbing aerosols, identifying the importance of implementing realistic aerosol optical characteristics, whenever available. Overall, our study finds no significant changes in precipitation amount over the largest part of the domain with the use of

realistic aerosol optical properties. Finally, the two simulations incorporating aerosol-cloud interactions present reduced liquid

water path and cloud fraction amounts compared to the control experiment that are mainly attributed to the change of the cloud microphysics scheme.

*Code and data availability.* The source code of the Weather Research and Forecasting Model (WRF) is freely available by UCAR/NCAR (http://www2.mmm.ucar.edu/wrf/users/downloads.html). The satellite data used (SARAH Edition1, CLARA-A1) are provided by EUMET-
SAT through the Satellite Application Facility on Climate Monitoring (CM SAF) (www.cmsaf.eu). The E-OBS gridded data set is provided by ECA&D project (http://www.ecad.eu). The MACv1 aerosol climatology data can be found at ftp://ftp-projects.zmaw.de. The ERA-Interim reanalysis and MACC aerosol data are available by the European Centre for Medium-Range Weather Forecasts (ECMWF) (https://apps.ecmwf.int/datasets/).

*Author contributions.* VP and EK designed the research. VP performed the experiments and analyzed the data. SK provided technical
assistance to the experiments. VP wrote the paper with inputs from all coauthors.

*Competing interests.* The authors declare that they have no conflict of interest

*Acknowledgements.* This research is co-financed by Greece and the European Union (European Social Fund- ESF) through the Operational Programme «Human Resources Development, Education and Lifelong Learning» in the context of the project "Strengthening Human Resources Research Potential via Doctorate Research" (MIS-5000432), implemented by the State Scholarships Foundation (IKY). We also
acknowledge the support of the Greek Research and Technology Network (GRNET) High Performance Computing (HPC) infrastructure for providing the computational resources necessary for the model simulations (pr006005_thin). NCAR is sponsored by the National Science Foundation. We acknowledge EUMETSAT for providing the satellite data through the Satellite Application Facility on Climate Monitoring (CM SAF) (www.cmsaf.eu).Furthermore we acknowledge the E-OBS dataset from the EU-FP6 project ENSEMBLES (http://ensembles-eu.metoffice.com) and the data providers in the ECA&D project (http://www.ecad.eu) and the use of MACv1 aerosol climatology data
(ftp://ftp-projects.zmaw.de). We also acknowledge ECMWF (www.ecmwf.int) for the provision of ERA-Interim reanalysis data as well as the MACC aerosol data.

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
