# Peer review of "Investigating the sensitivity to resolving aerosol interactions in downscaling regional model experiments with WRFv3.8.1 over Europe"

_Geoscientific Model Development, 2019_

## Referee Comment (RC1) · Anonymous Referee #1 · 16 Oct 2019

The submitted manuscript it performing sensitivity experiments following the EURO-CORDEX framework at 50km horizontal resolutions, where a series of 5-year simulations is performed with different treatments of aerosol. This is an important contribution to the regional climate community, as the approach of including aerosols in RCMs has not been well coordinated. However, before the current study can be published, the manuscript needs substantial improvements, specifically, in the understanding of the results, and also in the way the results are presented.

It is not clear for me the different between all the experiments listed in Table 1 (and section 2.4) (especially ARI_Mv1 ,ARI_Mv1urban, ARI_Mvfull), and this makes also

the results a bit difficult to understand. For someone who is not using WRF, would it be possible to explain this in a more general way, how the AOD is distributed into the different types, and what is the consequence of having rural over urban (now it is only written the percentage for different types, but it is not described in detail how this work). Some more information about the vertical distribution would also be good.

The results is a bit challenging to read, as there are many acronyms, and also it is refereeing to many figures in the supplementary text, which probably could have been included in the main text, and maybe some figures could be removed, as there are not always so large differences on the horizontal maps (I find figure S5 very useful, would it be possible to replace some of the tables and figures with this type of figure?).

Would it make sense to first present the results where the experiment CON, ARI-T, ARI_Mv1, ARI_Mv1urban, ARI_Mv1full is described, since this is focusing on the more trivial approach to include aerosol in RCMs, and it is just depending on what is having the "best" representation of the different species to represent the direct and semi-direct effect. Then a separate section can be presented, where it is shown the effect of including the aerosol-cloud interaction, which is representing the indirect effect. Then these two simulations (ACI and ARCI) can be compared with the "best" aerosol-representation from the first part (one of the CON, ARI-T, ARI_Mv1, ARI_Mv1urban, ARI_Mv1full). Now it is a lot of jumping back and forth between the different simulations, and it is not so easy to follow.

After reading the manuscript, I am not so sure what is the recommendation from the study, since when there is no aerosol climatology included (as in the CON-experiment), there is a cold bias over Europe, and this cold bias is enhanced when the aerosol is included (e.g. for ARI_T, ARI_Mv1). The ACI and ARCI simulations are warmer than the CON, so there is a potential to remove the cold bias when aerosol-cloud-interaction is included, but is this the take-home message that RCMs should aim for having interactive aerosol schemes? However, the impact on precipitation is very small, so in the end the aerosol treatment does not have a large impact?

If I have understood, there is no yearly change in the aerosol (only monthly or daily data), did the authors consider to include yearly varying aerosol? I guess for 5 years of simulation, the effect is not so large, but in past studies it has been shown that RCMs that don't have transient aerosol is not representing the change in the surface radiation correctly. (Bartok et al. (2017)])

General comments:

Line 15-17 (p1): "statistical significant ".what is statistical significant (and what is meant by "in some cases".. please rephrase.

Line 5-6 (p2): This sentence is not so easy to understand, especially if you don't know the difference between radiative forcing and adjustment (where I assume you mean rapid adjustment). Line 20-25 (p2 models and in EURO-CORDEX regional climate models for Europe Clim. Dyn. 49 2665–83 ] Line 15 (p5): I don't quite understand how the vertical profile of the different component is distributed in each model level. Is there some weighting doing the distribution? Would it be possible to show the vertical distribution for the different experiment?

Line 21-24 (p5): The distinguishing of rural, urban and maritime component is not clear for me. What is meant that "in this work the first two component has been implemented"? is the maritime not used? And for the experiment where the different components are used (e.g. rural or urban), is this the case for the whole domain? Or can you combine this and set rural for one part, and urban for another part? Line 2-4 (p12): are you describing a specific figure, or just the results in general? Line 9-10 (p12): does this mean that the model performance is actually better when aerosol is not included in the simulations? Line 7-9 (p13): is this related to a specific figure? Line 15 (fp13) From this line, it seems as a more general summary about the results is given, so it should maybe not be under section 3.2.4 (which is about the SW). Figure2: how about including S1 with Fig 2? Moreover, if possible, how about using a color scale which is white in the middle? (not green). Line 9 (p28): I would be careful with

using the word climate (e.g. "aerosol effect on European climate"), since only 5 years of simulation is done.

---

## Referee Comment (RC2) · Anonymous Referee #2 · 22 Oct 2019

**Review of the manuscript entitled** *Investigating the sensitivity to resolving aerosol interactions in downscaling regional model experiments with WRFv3.8.1 over Europe*, **by V. Pavlidis et al.**

This manuscript, submitted to *Geoscientific Model Development*, presents a sensitivity study about the impact of the aerosol representation in a regional model on regional climate. The objective of the study is to evaluate different regional simulations carried out with the WRF regional model using different aerosol configurations, and to show the consequence of these options on radiative budget, temperature, clouds and precipitation. Several observation datasets (E-OBS, CMSAF radiation data) are used for this evaluation.
Results show a strong impact of aerosols on surface shortwave radiation in all simulations compared against a control simulation without aerosols. Their radiative forcing is negative at the surface due to direct aerosol effect, only partially counteracted by a slight positive semi-direct effect. Temperature is therefore also reduced by aerosols, between -0.1 and 0.5°C, but no significant effect has been found for precipitation. Aerosol-cloud interactions modulate this impact on radiation. All these results are interesting and provide important conclusions for regional climate modelers, and in particular the WRF community. However, the organization of the paper is not very clear and not very easy to understand, and the following essential comments need to be corrected before considering a publication in GMD.

**Main comments:**
- The authors present eight different simulations (1 control run and 7 sensitivity experiments) which makes the paper difficult to understand, and the reader can easily get lost in the tables and figures presenting all the simulations. Besides, the author mainly focus only on ARI_T, ARI_Mv1urban, ACI and ARCI. The three other simulations (ARI_Mv1, ARI_Mv1full and ARI_MC) are not discussed in detail. Unless adding more discussions on these different simulations, I would suggest to keep only a few of them in the main text and in the tables and figures, and keep the other ones for supplementary material.

- The organization of the paper and in particular of Section 3.3 dealing with the sensitivty experiments should be improved. Indeed, the author present first temperature and precipitation, whereas the direct effect of aerosols concerns first radiation, which then has consequences on temperature. The fact to present cloud fraction at the end may also be a problem as this parameter is needed to explain aerosol effects on temperature and precipitation. I would suggest to reorganize Section 3.3, and notably start by the analysis on radiation.

- The simulations presented in this paper last only 5 years. I wonder if this is enough to study the sensitivity of climate-aerosol interactions, notably as far as cloud-aerosol interactions are concerned. I get the impression on some figures that the signal is quite noisy, notably in terms of cloud cover and precipitation.

- The presentation of figures should also be improved. Several figures (for example Figures 3 to 6) are composed of too many plots, which make them difficult to read, and not all of them are discussed in the paper. There are also too many references to figures in supplementary material. Some of them have their place in the main paper. Besides, the font used for labels should be higher (notably in Figures 1 and 2).

**Specific comments :**
- Abstract: It should be clearly stated in the abstract how are calculated the different numbers which are given, in particular the fact that they rely on a comparison with a control simulation without aerosol scattering and absorption.

- Page 2 Line 17-18: "Finally, a minority of the simulations use prognostic aerosol schemes with natural and anthropogenic emissions (dust, sea salt) online driven by meteorology". Could you precise which model has a fully prognostic aerosol scheme ? I don't see any model in the table given in footnote 1.

- Page 2 Line 20: "the aerosol-cloud interactions (indirect aerosol effect) is typically not considered". This information is not given in table in footnote1. Could you justify this point ?

- Page 2 Line 4: Could you precise here the version of WRF that you use ?

- Page 3 Line 13 (and Page 4 Line 19): the limits of the EURO-CORDEX domain given here (25S-75N, 40W-75E) seem to be very large for Europe (in particular 25S and 40W).

- Section 2.1.1: Please give the horizontal resolution of the observation datasets.

- Page 3 Line 16: What are these cases with an excess of 100% ? It is worth knowing if there is specific situations in which the E-OBS precipitation is not trustworthy.

- Page 3 Line 20: Please give a definition for Direct Normalized Irradiance.

- Section 2.1.2: Please give a reference for the SARAH dataset.

- Page 3 Line 24: "between ± 65° longitude and ± 65° latitude". Too large domain ?

- Page 4 Line 5: As the CLARA dataset is not used for the evaluation of radiation but only for cloud cover, it should be discussed if this could have an impact on the evaluation.

- Section 2.2: Please explain more clearly which indirect aerosol effects are taken into account in the different simulations (Twomey, Albrecht, …).

- Sections 2.3.1 and 2.3.2: The titles of these sections are unclear, they should be clarified.

- Page 5 Lines 14-15: In the case aer_opt=1, how are other radiative properties (SSA, asymmetry parameter) defined ? Are there common for all aerosol types ?

- Page 6 Lines 18-23: It is not clear for me how this aerosol scheme is used. Is it a full prognostic aerosol scheme with emissions, transport and deposition ?

- Page 7 Lines 6 and 14: what does mp=8 (and mp=28) mean ?

- Page 7 Lines 8-9: "The single scattering albedo (SSA) at 550nm of the ''rural" type aerosols ranges in our experiments between 0.92 and 0.98". How are these values spatially distributed ? Maybe a map in supplementary material could be helpful to understand the spatial distribution of aerosol-radiation effects.

- Table 1: For simulation ACI, aerosols are indicated to interact with clouds whereas the option aer_opt=0 is used. How is it possible ?

- Page 9 equation2: The DRE is calculated in clear-sky fluxes, so I suggest to call it rather CDRE (Clear-sky Direct Radiation Effect) to avoid confusion with RE, which is calculated in all-sky confitions.

- Page 10 Line 13: "The climatology of Tegen has a lower AOD compared to SEVIRI, but follows the latter's seasonal spatial variability". I don't understand how the Tegen AOD follows the seasonal spatial variability of SEVIRI AOD, it is not clear from me in Figure 1.

- Page 10 Line 15: The use of AOD assimilation in the MACC reanalysis could be mentioned to explain the better agreement of MACC AOD with satellite data.

- Figure 1: There is strange high AOD in Eastern Europe in SEVIRI data in winter (DJF). Please comment on this pattern. Maybe the use of another satellite product (MODIS for example) could help to ensure the robustness of satellite data.

- Page 12 Lines 2-4: These lines should be rather in the conclusion of the section than at the beginning. The authors could rather introduce Figure 2.

- Page 12 Lines 6-7: There is on the contrary a warm bias in northern Scandinavia.

- Page 12 Line 10: Please give a reference for the Noah land surface model.

- Page 12 Line 25: Please give a reference for the WDM6 cloud microphysics scheme.

- Section 3.2.3: The bias in cloud fraction in summer could be related to a too zonal circulation ? Besides, the author could discuss if those biases in cloud cover could have an impact on AOD in the case of the simulations with aerosol-cloud interactions.

- Tables 2, 3 and 4: Please explain which domain is used for these averages. In particular, it should be stated if only land points are considered (as the model is not coupled with ocean, it would be more relevant to show only land grid points).

- Page 15 Line 7: The impact on surface temperature seems to be larger in autumn than in summer, while the AOD is higher in summer. Is there a role of internal variability ?

- Page 15 Line 9: "in cases reaching a decrease of 1.5° C" Please give more details on these cases.

- Page 15 Lines 10-12: This point should be related to a figure.

- Page 15 Lines 13-end: The results using the ARI_Mv1urban simulation should be moderated as the absorption of aerosols is not realist in this simulation.

- Page 16 Line 5: Could you explain: "Contrary to the ARI group, simulation ACI using the Thompson aerosol-cloud interacting cloud microphysics and accounting for indirect effects only results in a domain averaged temperature increase (0.1 to 0.2 o C) compared to CON for all seasons except autumn" ?

- Page 16 Line 21: It should be mentioned that the Black Sea is not coupled, which could influence the results on precipitation.

- Page 17 Lines 2-4: This point should be related to a figure.

- Page 17 Lines 12-13: how is calculated the correlation with AOD in the ARCI simulation (with which AOD) ? The difference in AOD between ARCI and ACI could have an impact ?

- Page 19 Line 14: it is difficult to draw this conclusion as AOD is not the same in all simulations. I

suggest that AOD could be added in Table 4 to discuss this point.

- Page 19 Lines 21-24: I don't understand how the author come to this conclusion.

- Page 20 Line 1: what is Aer2urban ?

- Page 20 Line 8: Could you explain why the effects are stronger on DNI than on Rsds ?

- Page 21 Line 5: what is the specific effect of aerosols on diffuse radiation, that could be distinguished from direct radiation ?

- Page 23 Lines 11-12. This result involving LWP seems to be important to understand cloud-aerosol interactions. Please explain more this process.

- Page 23 Lines 19-20. In this kind of semi-direct effect, the internal model variability could be important. The author should mention this point.

- Page 26 Lines 26-27. I don't understand how "the introduction of aerosol-radiation interactions" could lead to "more transparent clouds" ?

**Other corrections:**
- Abstract line13 and Page 19 line 6: is comprised of (instead of comprises of)
- Page 4 Line 10: an underestimation
- Page 4 Line 30: incorporates aerosol**s**
- Page 10 Line 15: The MACC reanalysis is **in better agreement** with the satellite data.
- Page 10 Line 16: and the  generally higher AOD
- Figure 1: Please keep the same spelling for **MAC-v1**
- Page 12 Line 11: **In particular** northern Europe is …
- Page 13 Line 21:  compensation between errors
- Figure 4: Space character is missing after (RE)
- Page 28 Line 2: please define NWP

---

## Author Comment (AC1) · 17 Dec 2019

Author's Response

First of all I would like to thank both reviewers for their time reading the manuscript and for their meticulous comments, which have been very helpful and have contributed to the improvement of the manuscript. I responded to all comments and implemented the changes in a new manuscript, which is re-structured and more readable. Below, I provide a short overall description of the manuscript changes and then I proceed to the responses.

General changes to the manuscript

1. **Restructuring of the Results**
   The structure of the Results (section 3.3) has been altered. Instead of describing the impact on each variable separately we describe the results depending on the interactions enabled:
   a) Enabling only aerosol-radiation interactions
   b) Enabling aerosol-radiation interactions in an environment where the aerosol-cloud interactions are also present
   c) Comparing the Thompson aerosol aware microphysics scheme to the non-aerosol aware Thompson2008 scheme and describing what happens when we implement both the aerosol-aware scheme with aerosol-cloud interactions and aerosol-radiation interactions and compare against control simulation that has no aerosol effects at all.

2. **Adding figure, removing tables**
   A new figure (Fig2-in the new manuscript) has been added containing the vertical aerosol profiles of the simulations. Tables 3 and 5 (original manuscript)have been moved to the supplement. Furthermore a new figures has been added in the supplement describing the single scattering albedo of the simulations.

3. **Changing color scale and number of simulations depicted in the main figures**
   We have changed the color scale in the figures (2, 3,4,5,6-original manuscript). The new colorbar has white color in the middle and makes clearer the sign of the small changes. Moreover the number of simulations depicted has been reduced. I understand that the previous figures were too packed with information. Several simulations enabling aerosol-radiation interactions have very similar behavior and are not discussed thoroughly in the manuscript. Now instead of 8 rows of plots the new figures have only 5. We keep simulation ARI_T (Tegen climatology, 1[st] aerosol option), ARI_Mv1 (Macv1 climatology, 2[nd] aerosol option), ARI_Mv1urban (urban very absorbing aerosol), the difference ARCI-ACI (indicative of aerosol-radiation interaction when indirect effect is present) and ARCI. We have omitted simulations ARI_Mv1full, ARI_MC that have very similar behavior to the others enabling aerosol-radiation interaction and also simulation ACI since the depiction of ARCI is enough to state the thing we describe about the Thompson aerosol aware scheme.

Now I proceed to answer each reviewer's comments. The original comments are with bold fonts and each response lies below the respective comment.

**It is not clear for me the different between all the experiments listed in Table 1 (and section 2.4) (especially ARI_Mv1 ,ARI_Mv1urban, ARI_Mvfull)**

Answer: Simulations ARI_Mv1, ARI_Mv1urban and ARI_Mv1full have the same AOD field (MACv1) but they have differences in the rest aerosol optical properties (single scattering albedo, asymmetry factor).

Changes in the manuscript: I have added this phrase to make it more clear (page 8, line22 new manuscript): "Within the ARI group simulations ARI_Mv1, ARI_Mv1urban and ARI_Mv1full have the same AOD field (MACv1) but they have differences in the rest aerosol optical properties (single scattering albedo, asymmetry factor)."

**For someone who is not using WRF, would it be possible to explain this in a more general way, how the AOD is distributed into the different types, and what is the consequence of having rural over urban**

Answer: The consequence of using rural over urban is being described in section 2.4 Model Simulations, page 7, line 8 of the reviewed manuscript. I have included a new figure (S2) with single scattering albedo maps in the supplement that will make the rural over urban difference even clearer.

**Some more information about the vertical distribution would also be good.**

Answer: I have added an extra figure (fig2) in the manuscript regarding the vertical distribution and a small text (page 13):

[Figure]

**The results is a bit challenging to read, as there are many acronyms, and also it is refereeing to many figures in the supplementary text, which probably could have been included in the main text, and maybe some figures could be removed, as there are not always so large differences on the horizontal maps (I find figure S5 very useful, would it be possible to replace some of the tables and figures with this type of figure?).**

Answer: I believe the restructuring of the Results, described in the beginning, is helping make the results easier to understand. Less maps are included in the main figures. I think the reviewer is referring to figureS6 (not S5) in the supplement. I have included a figure with box plots to the supplement (S1) and moved tables 3 and 5 of the original manuscript to the supplement.

**Would it make sense to first present the results where the experiment CON, ARI-T, ARI_Mv1, ARI_Mv1urban, ARI_Mv1full is described, since this is focusing on the more trivial approach to include aerosol in RCMs, and it is just depending on what is having the "best" representation of the different species to represent the direct and semidirect effect. Then a separate section can be presented, where it is shown the effect of including the aerosol-cloud interaction, which is representing the indirect effect. Then these two simulations (ACI and ARCI) can be compared with the "best" aerosol representation from the first part (one of the CON, ARI-T, ARI_Mv1, ARI_Mv1urban,ARI_Mv1full). Now it is a lot of jumping back and forth between the different simulations, and it is not so easy to follow.**

Answer: The reviewer is suggesting an interesting way to restructure the section of the Results. The restructure described in the beginning is very similar. First only the aerosol-radiation interactions are presented, the trivial approach to include aerosol in RCMs. Then we explore how aerosol-radiation interactions behave if we enable them in an environment where the indirect effect is also present. Finally what is the impact of the Thompson aerosol-aware scheme that enables aerosol-cloud interactions.

Two significant notes:

Simulation ARI_T uses the option aer_opt=1 to parameterize aerosol-radiation interactions. Simulations ARI_Mv1, ARI_Mv1full, ARI_Mv1urban and ARI_MC use the option aer_opt=2. Thus the impact of aerosol-radiation interactions is not just depending on the aerosol dataset used but also to the option (parameterization) used. This is why the Tegen climatology used in ARI_T leads to a similar clear-sky shortwave radiation decrease with ARI_Mv1 (for example) despite the fact that the AOD of Tegen is considerably smaller tthan that of MACv1. The aer_opt=1 parameterization has a tendency to decrease clear-sky radiation more per unit of AOD ($W/m^2$/AOD) than aer_opt=2.

When using the Thompson aerosol aware microphysics scheme, for example in ACI, we do not just implement aerosol-cloud interactions. We introduce a modified microphysics scheme that is designed to work with aerosol. Thus when we compare ACI to control CON (that uses the Thompson2008 non-aerosol microphysics) the impact seen is attributed to the

combined effect of the introduction of a different microphysics scheme and to the possible indirect effect taking place.

**After reading the manuscript, I am not so sure what is the recommendation from the study, since when there is no aerosol climatology included (as in the CON-experiment), there is a cold bias over Europe, and this cold bias is enhanced when the aerosol is included (e.g. for ARI_T, ARI_Mv1). The ACI and ARCI simulations are warmer than the CON, so there is a potential to remove the cold bias when aerosol-cloud interaction is included, but is this the take-home message that RCMs should aim for having interactive aerosol schemes? However, the impact on precipitation is very small, so in the end the aerosol treatment does not have a large impact?**

Answer: I am not very worried about whether the aerosol introduction improves or worsens the bias. Of course if aerosol inclusion increased dramatically the bias this would be alarming and an indication that the aerosol parameterizations are probably not working properly. But this is not the case in our study. The final bias is of course a product of many different things such as the RCM structure and setup and the quality of the driving data. The purpose of our study is to identify the impact of the aerosol parameterizations and data used and the general behavior of aerosol implementation in the WRF model over Europe. This impact can improve or worsen the bias depending on the characteristics of each simulation. It is therefore simulation specific. For example the use of a different land scheme in our study could lead to a warm bias in the control simulation and thus enabling aerosol-radiation interactions would end up improving the temperature bias. So it is up to each WRF user to decide whether and what options to enable if they have bias improvement in mind. I can only describe their impact. Therefore I do not want to make general recommendations about using this or the other option or dataset. The same goes for the ACI and ARCI simulations that are mentioned by the reviewer. ACI is indeed slightly warmer than control and can improve the cold bias. But I would not state that it is a take-home message that RCMs should aim for having interactive aerosol schemes only because of the bias improvement. Using a different model setup, different driving data or a different domain could lead to bias increase when using interactive aerosol. It is understandable that people may want to include in their simulations all the physical mechanisms that are available and have a more detailed representation of each phenomenon (e.g. interactive aerosol). This makes the model more complete. However the extra or/and more detailed mechanism does not necessarily improve the bias. Regarding precipitation, indeed aerosol treatment does not seem to have a large impact. To conclude I can make two statements regarding the bias:

a) Aerosol introduction can have an impact on bias. However the main biases are not altered considerably. Thus aerosol do not seem to be the main source of the bias.

b) Aerosol introduction does not necessarily improve the bias. In many cases the bias is increased when aerosol are enabled in our study. This does not mean that the aerosol parameterizations have a problem. We have seen that they behave in a

physically consistent way. Whether bias is improved or not is specific to each simulation. It depends on each user to decide if and which aerosol treatment to use.

**If I have understood, there is no yearly change in the aerosol (only monthly or daily data), did the authors consider to include yearly varying aerosol? I guess for 5 years of simulation, the effect is not so large, but in past studies it has been shown that RCMs that don't have transient aerosol is not representing the change in the surface radiation correctly. (Bartok et al. (2017) )**

Answer: The Tegen and MACv1 climatologies do not have yearly variability, only monthly variations. The MACC is a daily dataset and thus there are differences from year to year. However the impact of this year to year variability is minimal. It seems that 5years are indeed a small period to explore the effect of transient aerosol and this was not the intention of this study. It is a very nice suggestion by the reviewer to explore the effect of transient aerosol. Actually we are working on longer simulations spanning 30 years that indeed show that the inclusion of transient aerosol is important to correctly capture the trends in surface radiation. However this will be the subject of another study.

**General comments**:

**Line 15-17 (p1): "statistical significant ".what is statistical significant (and what is meant by "in some cases"..please rephrase.**

Answer: We use the phrase "statistically significant" to denote that a result is of statistical significance. As explained in the methodology we use the Mann-Whitney non-parametric test and calculate significance at the 0.05 level. I am under the impression that the phase "statistically significant result" is frequently used.

I have rephrased to: "of statistical significance".

The use of the phrase "in some cases" was meant to state that the changes that are of statistical significance are not widespread but are seen in some selective cases. I did not want to give more details in the abstract, just state the general behavior. I have added: "which in some cases, mainly close to the Black Sea in autumn".

**Line 5-6 (p2): This sentence is not so easy to understand, especially if you don't know the difference between radiative forcing and adjustment (where I assume you mean rapid adjustment).**

Answer: Yes indeed we mean the rapid adjustments. I do understand that it is hard to follow. I think it is better to keep it simple and just state that both aerosol-radiation and

aerosol-cloud interactions add to the uncertainty while the aerosol-cloud interactions do that to a larger degree.

Rephrased to: "It states that the uncertainty due to aerosol is attributed to both aerosol-radiation (ari) and aerosol-cloud interactions (aci) with the latter having the largest contribution."

**Line 15 (p5): I don't quite understand how the vertical profile of the different component is distributed in each model level. Is there some weighting doing the distribution? Would it be possible to show the vertical distribution for the different experiment?**

Answer: I have added a figure depicting the vertical profile of the experiments in the supplement (S1). The option aer_opt=1 (ARI_T) uses the 3D Tegen climatology and constructs a vertical profile by adding the AOD of each aerosol type in each model level. The option aer_opt=2 (ARI_Mv1 family) assumes a specific function for the vertical profile. This is the same over each grid point and thus the vertical profile has the same shape over each grid point, only the magnitude of AOD changes depending on the total AOD provided at each grid point.

**Line 21-24 (p5): The distinguishing of rural, urban and maritime component is not clear for me. What is meant that "in this work the first two component has been implemented"? is the maritime not used? And for the experiment where the different components are used (e.g. rural or urban), is this the case for the whole domain? Or can you combine this and set rural for one part, and urban for another part?**

Answer: The "rural", "urban" and "maritime" types are different ways that the option aer_opt=2 uses to parameterize the single scattering albedo (SSA) and angstrom exponent (AE) of aerosol. The selected type is indeed used for the entire domain and you cannot chose one type for one part of the domain and another type for the another part. However for the selected aerosol type the produced fields of SSA and AE are not spatially homogenous but present spatial variability over the domain. This is because the parameterization takes into account the total AOD over each grid point and the relative humidity to calculate the SSA and AE values. The difference between the types lies mainly in how absorbing are the aerosols of each type. The range of the single scattering albedo is given in the manuscript in section 2.4 "Model Simulations". I have also included a figure depicting the SSA of the simulations after a request of the second reviewer. The main difference of the "rural" and "urban" type is that "urban" is considerably more absorbing all over the domain. Indeed we only used the "rural" and "urban" types and not "maritime". "Maritime" presents larger scattering compared to "rural" however the differences are not as large as those between "rural" and "urban". Since we tested the "rural" type which is the most balanced and also used the more realistic SSA of MACv1 in one simulation (MACv1full) we believe that the use of "maritime" would not provide any significant additional value and some early tests gave the same indication. We chose however to test the "urban" type since it represents a completely different situation from the "rural" and "maritime" types and would show the impact of extremely absorbing aerosol, especially the semi-direct effect.

I have added the phrase (page 6, line 7 new manuscript): "Only one aerosol "type" can be used for the entire domain."

**Line 2-4 (p12): are you describing a specific figure, or just the results in general?**

Answer: This is indeed a small summary of the overall evaluation. I think it is nice to clearly state the main point in the beginning and then elaborate for each variable.

**Line 9-10 (p12): does this mean that the model performance is actually better when aerosol is not included in the simulations?**

Answer: Yes in some cases, not only temperature, model results present smaller biases when aerosols are not included. As I have described in a previous answer I do not think that this is problematic regarding the performance of the aerosol parameterizations used. The fact that the inclusion of an additional physical mechanism in the model (aerosol) fails to improve the bias is quite interesting and reveals how models work. That they have after all a group of different parameterizations that do not always work perfectly with each other and in many cases these parameterizations are calibrated to work well at certain conditions. For example a radiation parameterization might be calibrated to produce small biases without the use of a separate aerosol parameterization in place. IN this case the aerosol impact is indirectly taken into account through the calibration process. However this calibration might be insufficient in another domain (e.g larger AOD values) under different meteorological conditions (e.g. heavy cloudiness) or different aerosol related events (e.g transportation of very absorbing aerosol).

**Line 7-9 (p13): is this related to a specific figure?**

Answer: Yes this is related to figure 3 (fig2 original manuscript). It has been added to the manuscript.

**Line 15 (fp13) From this line, it seems as a more general summary about the results is given, so it should maybe not be under section 3.2.4 (which is about the SW).**

Answer: Yes this is a very helpful comment. This is a general summary about the evaluation of the aerosol including sensitivities. It is under a separate section after finishing with the evaluation of the control simulation.

I have moved this text in a new section (3.2.5 Evaluation of the sensitivity simulations) to make clear we are talking about the sensitivity simulation evaluation.

**Figure2: how about including S1 with Fig 2? Moreover, if possible, how about using a color scale which is white in the middle? (not green).**

Answer: I understand this point. However the differences in the bias are small (except only for DNI) and there are several maps in the manuscript and supplement explaining the impact of aerosol on each variable. I would prefer to just have this figure alone to give emphasis on the evaluation of the control simulation. I have changed the color scale to a new one having white in the middle.

**Line 9 (p28): I would be careful with using the word climate (e.g. "aerosol effect on European climate"), since only 5 years of simulation is done.**

Answer: This is indeed true. 5 years are not a sufficient period to state impact on climate. I have replaced "on European climate" to "over Europe".

Reviewer 2

**Main comments:**

**- The authors present eight different simulations (1 control run and 7 sensitivity experiments) which makes the paper difficult to understand, and the reader can easily get lost in the tables and figures presenting all the simulations. Besides, the author mainly focus only on ARI_T, ARI_Mv1urban, ACI and ARCI. The three other simulations (ARI_Mv1, ARI_Mv1full and ARI_MC) are not discussed in detail. Unless adding more discussions on these different simulations, I would suggest to keep only a few of them in the main text and in the tables and figures, and keep the other ones for supplementary material.**

**- The organization of the paper and in particular of Section 3.3 dealing with the sensitivity experiments should be improved. Indeed, the author present first temperature and precipitation, whereas the direct effect of aerosols concerns first radiation, which then has consequences on temperature. The fact to present cloud fraction at the end may also be a problem as this parameter is needed to explain aerosol effects on temperature and precipitation. I would suggest to reorganize Section 3.3, and notably start by the analysis on radiation.**

Answer: I do understand the comments regarding the organization off the results and the general readability of the entire manuscript and consider them extremely important since the essence of a paper is to easily communicate information. I have tried to reorganize the entire manuscript and especially the way Results are presented by implementing suggestions by the two reviewers. The main changes are described in the beginning of the author's reply. To quickly summarize they include two key features: a)The aerosol impact is presented according to the effect explored and not for each variable separately, b) less simulations depicted in the figures and tables 3 and 5 have been moved to the supplement.

**- The simulations presented in this paper last only 5 years. I wonder if this is enough to study the sensitivity of climate-aerosol interactions, notably as far as cloud-aerosol interactions are concerned. I get the impression on some figures that the signal is quite noisy, notably in terms of cloud cover and precipitation.**

Answer: Very interesting question. I do believe that 5 years are enough to study the rapid climate responses (or adjustments) due to the forcing of aerosol. According to the IPCC AR5 (section 7.1.3) most of the rapid adjustments are thought to occur within few weeks. The slow adjustments (mainly the full extent of ocean atmosphere interaction) only need much larger simulation time. Other RCM studies have produced results for smaller or similar periods: Da Silva 2018-6 moths for precipitation, Zanis 2009-2 years, Nabat 2015-2 years spin up +6 years simulations. I am not very worried about the somewhat noisy signal in precipitation and cloud fraction since these variables are highly impacted by differences in the local scale especially terrain elevation.

Zanis, P.: A study on the direct effect of anthropogenic aerosols on near surface air temperature over Southeastern Europe during summer 2000 based on regional climate modeling, Annales Geophysicae, 27, 3977–3988, https://doi.org/10.5194/angeo-27-3977-2009, http://www.ann-geophys.net/27/3977/2009/, 2009.

Da Silva, N., Mailler, S., and Drobinski, P.: Aerosol indirect effects on summer precipitation in a regional climate model for the Euro-Mediterranean region, Annales Geophysicae, 36, 321–335, https://doi.org/10.5194/angeo-36-321-2018, https://www.ann-geophys.net/36/321/2018/, 2018.

Nabat, P., Somot, S., Mallet, M., Sevault, F., Chiacchio, M., andWild, M.: Direct and semi-direct aerosol radiative effect on the Mediterranean climate variability using a coupled regional climate system model, Climate Dynamics, 44, 1127–1155, https://doi.org/10.1007/s00382-014-2205-6, 2015.

**- The presentation of figures should also be improved. Several figures (for example Figures 3 to 6) are composed of too many plots, which make them difficult to read, and not all of them are discussed in the paper. There are also too many references to figures in supplementary material. Some of them have their place in the main paper. Besides, the font used for labels should be higher (notably in Figures 1 and 2).**

Answer: The number of the plots in the main figures has been reduced. Unfortunatelly it is difficult to avoid referencing the supplement a few times. The fonts in Figures 1 and 3 (fig2 of original manuscript) have been placed slightly higher.

**Specific comments :**

**- Abstract: It should be clearly stated in the abstract how are calculated the different numbers which are given, in particular the fact that they rely on a comparison with a control simulation without aerosol scattering and absorption.**

Answer: I added the phrase (page 1, line19 new manuscript): "The impact of aerosol is calculated by comparing against a simulation that has no aerosol effects."

**- Page 2 Line 17-18: "Finally, a minority of the simulations use prognostic aerosol schemes with natural and anthropogenic emissions (dust, sea salt) online driven by meteorology". Could you precise which model has a fully prognostic aerosol scheme ? I don't see any model in the table given in footnote 1.**

Answer: Only one model uses a prognostic scheme. It is UM-WRF361 (third from the bottom in the table). It states that aerosol are estimated online at every time step. I have rephrased to state that it is only one model since the term "a minority" is not precise.

Changed to: "Finally only one model uses a prognostic aerosol scheme estimating online the aerosol field."

**- Page 2 Line 20: "the aerosol-cloud interactions (indirect aerosol effect) is typically not considered". This information is not given in table in footnote1. Could you justify this point ?**

Answer: Indeed this is a good point. The fact that aerosol-cloud interactions are not considered was indirectly inferred by the table and a bit hasty. There is only one model (SMHI-RCA4) that states clearly that has only aerosol-radiation interactions (and of course 5 cases that have no aerosol at all (so no indirect effect) and a couple that are not sure). Most of the models however use climatologies or fixed AOD fields and usually in these cases this indicates simple aerosol sophistication and only aerosol interaction with radiation. Moreover even if aerosol-cloud interactions are considered I do not believe that the full impact of the indirect effect could be captured with a stable in time aerosol field (even with seasonal variability). Only a prognostic aerosol scheme can modify the aerosol field according to the meteorological conditions and thus fully capture the impact

But yes, the point I am trying to make here cannot be strictly justified by the information given in the table in footnote1. This statement has been omitted in the new manuscript.

**- Page 2 Line 4: Could you precise here the version of WRF that you use ?**

Answer: I have added: "WRFv3.8.1"

**- Page 3 Line 13 (and Page 4 Line 19): the limits of the EURO-CORDEX domain given here (25S- 75N, 40W-75E) seem to be very large for Europe (in particular 25S and 40W).**

Answer: Nice observation. There is a typo in the description of the domain and yes this is not the typical EURO-CORDEX domain but the domain used in the simulations. The southern limit is 25N and not 25S. It has been corrected in the manuscript in the second case stated above. In the first case (section 2.1.1) I have rephrased to simply state that E-OBS cover Europe. The 40W and 75E are not as distanced from the original EURO-CORDEX domain as they seem. Since we use a rotated grid only the domain upper left and right corners stretch to such limits. The domain used in the simulations encompasses completely the EURO-CORDEX domain.

**- Section 2.1.1: Please give the horizontal resolution of the observation datasets.**

Answer: It is actually stated that data on a $0.44^{o}$ rotated pole grid are used.

**- Page 3 Line 16: What are these cases with an excess of 100% ? It is worth knowing if there is specific situations in which the E-OBS precipitation is not trustworthy.**

Answer: The largest relative errors in precipitation are found mainly over mountainous areas in the Alps and mountainous parts of Norway, in North Africa due to the very small station density and in areas east to the Baltic sea. Large relative errors are also seen in some grid cells in Italy and Spain. In general results are the best over central Europe and the UK. It must be noted that these "errors" are estimated by comparison against more regional gridded datasets with higher density of stations that are thought to be closer to reality. I have rephrased the sentence in order to be more specific and provide as much information is given in Hofstra et al. (2009) as possible.

Hofstra, N., Haylock, M., New, M., and Jones, P. D.: Testing E-OBS European high-resolution gridded data set of daily precipitation and surface temperature, Journal of Geophysical Research Atmospheres, 114, https://doi.org/10.1029/2009JD011799, 2009.

Rephrased to: When compared against regional datasets with higher station density (Hofstra et al., 2009) the E-OBS dataset presented a mean absolute error around $0.5^{o}C$ for temperature whereas for precipitation a general tendency of underestimating precipitation amount is reported, with large (>75%) relative errors found in mountainous regions of the Alps and Norway, over North Africa and in areas east to the Baltic Sea.

**- Page 3 Line 20: Please give a definition for Direct Normalized Irradiance.**

Answer: I have added: DNI is the solar radiation received by the direction of the sun's rays and received by a surface that is perpendicular to that direction.

**- Section 2.1.2: Please give a reference for the SARAH dataset.**

Answer: A reference is actually given in the second line: (Müller et al., 2015). I have moved it to the first line in order to state it right away.

**- Page 3 Line 24: "between ± 65° longitude and ± 65° latitude". Too large domain ?**

Answer: I have checked this again. It is stated in the CMSAF webpage https://wui.cmsaf.eu/safira/action/viewDoiDetails?acronym=SARAH_V002 and it is actually true regarding the ultimate extent of the domain. There are variations depending on season.

**- Page 4 Line 5: As the CLARA dataset is not used for the evaluation of radiation but only for cloud cover, it should be discussed if this could have an impact on the evaluation.**

Answer: I understand your point. Every dataset uses its own information about cloudiness amount to calculate shortwave radiation at the surface. However the use of a different product for radiation and cloud fraction does not necessarily lead to discrepancies as long as these products are good enough in estimating the respective variable. Both SARAH for Rsds (Müller et al., 2015) and CLARA for cloud fraction (Karlsson and Hollmann, 2012) have reasonable accuracy in detecting the respective variable, so I do not believe this has a significant impact on evaluation results.

I have added the phrase (page 4, line27):  "The use of a different product for cloud fraction (CLARA) than the one used for radiation (SARAH) does not impact the evaluation since both of these products have reasonable accuracy and uncertainty in estimating the respective variables."

**- Section 2.2: Please explain more clearly which indirect aerosol effects are taken into account in the different simulations (Twomey, Albrecht, …).**

Answer: I have added the phrase: "Thus aerosols are free to either change cloud albedo (first indirect or Twomey effect) or/and impact cloud lifetime (second or Albrecht indirect effect)."

**- Sections 2.3.1 and 2.3.2: The titles of these sections are unclear, they should be clarified.**

Answer: The section 2.3.1 title was changes from: "WRF Aerosol options" to "WRF aerosol parameterizations examined".
The section 2.3.2 title was changed from: "Aerosol data" to "Aerosol datasets used".

**- Page 5 Lines 14-15: In the case aer_opt=1, how are other radiative properties (SSA, asymmetry parameter) defined ? Are there common for all aerosol types ?**

Answer: In aer_opt=1 the other radiative properties (SSA,ASY,) are given for each aerosol type separately in lookup tables the model. Then for each grid cell, each model level and each spectral band of the radiation scheme a final value is calculated by weighing the value of each aerosol type by its AOD and then summing them all together.

I have added the phrase (page 6, line 11):  "The single scattering albedo and asymmetry factor are given for each aerosol type and a final value is calculated in each model level and for each spectral band of the radiation scheme. This is done by weighting the value of each aerosol type by its respective AOD and aggregating for all five aerosol types."

**- Page 6 Lines 18-23: It is not clear for me how this aerosol scheme is used. Is it a full prognostic aerosol scheme with emissions, transport and deposition ?**

Answer: The Thompson aerosol-aware scheme is a very interesting idea. It is a microphysics scheme that is prognostic in the sense that it explicitly predicts the number concentration of aerosols and it has emissions, transport and deposition for aerosol. It makes some simplifications (like all schemes do to some extent). It separates aerosol into two general species: droplet nucleating and ice nucleating. It uses an aerosol climatology, derived from multi-year (2001-2007) global model simulations by the Goddard Chemistry Aerosol Radiation and Transport (GOCART) model, to initialize the aerosol field in the model and provide boundary conditions. Moreover a fake surface aerosol emissions/flux/tendency is added at the surface for the droplet-nucleating aerosol. This emission flux is based on near-surface aerosol concentration and a simple mean surface wind.
To conclude: The Thompson aerosol-aware scheme is not as complex as schemes with interactive chemistry that contain features like multiple aerosol types, multiples bin sizes, realistic emissions inventory e.t.c. It is a microphysics scheme that tries to have a more simplified aerosol treatment that has no significant extra computational burden. Its purpose is not to predict aerosol concentrations with the best accuracy. It tries to retain some basic climatological features of aerosol while at the same time changing the aerosol field according to the meteorological conditions thus enhancing the realism of aerosol-cloud interactions.
I have added(page 6, line26):  "This scheme explicitly predicts aerosol number concentrations."

**- Page 7 Lines 6 and 14: what does (and mp=28) mean ?**

Answer: Indeed this is not explained. This is the symbolism of each microphysics option in the model namelist. I have added: (mp=8-in the model namelist)

**- Page 7 Lines 8-9: "The single scattering albedo (SSA) at 550nm of the "rural" type aerosols ranges in our experiments between 0.92 and 0.98". How are these values spatially distributed ? Maybe a map in supplementary material could be helpful to understand the spatial distribution of aerosol-radiation effects.**

Answer: I have added a map of the SSA values on the supplement (S2).

**- Table 1: For simulation ACI, aerosols are indicated to interact with clouds whereas the option aer_opt=0 is used. How is it possible ?**

Answer: I see this might be confusing. The "Aerosol option" row in Table1 is for aerosol-radiation interactions only. Aer_opt=0 means that there are no aerosol-radiation interactions. Thus ACI has aerosol-cloud interactions since it uses the Thompson aerosol-aware scheme but the aerosol of the Thompson scheme do not interact with radiation.

**- Page 9 equation2: The DRE is calculated in clear-sky fluxes, so I suggest to call it rather CDRE (Clear-sky Direct Radiation Effect) to avoid confusion with RE, which is calculated in all-sky confitions.**

Answer: I think this adds to the clarity of the metric. I have renamed it to Clear-sky Direct Radiation Effect  CDRE throughout the manuscript.

**- Page 10 Line 13: "The climatology of Tegen has a lower AOD compared to SEVIRI, but follows the latter's seasonal spatial variability". I don't understand how the Tegen AOD follows the seasonal spatial variability of SEVIRI AOD, it is not clear from me in Figure 1.**

 Answer:  This was meant to state that Tegen has the same main changes in seasonal AOD like SEVIRI.  I have completely re-written the section 3.1 Aerosol optical depth omitting this sentence. I tried to give more precise information. We have also included the MODIS satellite dataset and use it as the main point of reference. We reanalyzed the SEVIRI dataset taking more care about averaging missing values and

**- Page 10 Line 15: The use of AOD assimilation in the MACC reanalysis could be mentioned to explain the better agreement of MACC AOD with satellite data.**

Answer: I have included that MACC uses data assimilation is section 3.1 (page 11 ,line 15 )

**- Figure 1: There is strange high AOD in Eastern Europe in SEVIRI data in winter (DJF). Please comment on this pattern. Maybe the use of another satellite product (MODIS for example) could help to ensure the robustness of satellite data.**

Answer: I have also used the MODIS dataset and this is the main point of reference now. This weird AOD high over Eastern Europe in SEVIRI during winter severely confused me in the beginning. I was unable to find a reference of this. However this AOD high is fake and had to do with the way the missing values were seasonally averaged due to a bug in the code. The SEVIRI monthly mean gridded data have large areas of missing values in winter and autumn over around 45N. Few scattered grid points with valid data can be found however in higher latitudes in each month. By mistake the seasonal averaging happened also above grid points that had very few valid monthly (even one valid monthly values per the entire 5 year period was enough to be included). Thus the averaging over those areas resulted in an aerosol field that cannot be representative of the seasonal mean. In the corrected analysis only grid points that have no missing values at all are used for the seasonal averaging.

**- Page 12 Lines 2-4: These lines should be rather in the conclusion of the section than at the beginning. The authors could rather introduce Figure 2.**

Answer: I believe it is nice to have a quick summary in the beginning with the essence of the evaluation and then elaborate more for each variable.

**- Page 12 Lines 6-7: There is on the contrary a warm bias in northern Scandinavia.**

Answer: In Scandinavia over winter the averaged bias is negative and almost -1 $^{o}$C. I understand however since the warm bias at the north is not discussed in the manuscript the statement of considerable negative bias in Scandinavia might be confusing.
I have rephrased to make it clearer.

Changed to : In the simulation CON winter temperatures are mostly underestimated (-0.5oC domain average, land only), with higher cold biases over Scandinavia (despite a warm bias at the north), the Mediterranean and the Alps (-1$^{o}$C) as indicated in the upper panel of (Fig.2).

**- Page 12 Line 10: Please give a reference for the Noah land surface model.**

Answer: I have added the reference

Niu, G-Y., Yang, Z. L., Mitchell, K. E., Chen, F., Ek, M. B., Barlage, M., ... Xia, Y. (2011). The community Noah land surface model with multiparameterization options (Noah-MP): 1. Model description and evaluation with local-scale measurements. *Journal of Geophysical Research: Space Physics*, *116*(12), [D12109]. https://doi.org/10.1029/2010JD015139

**- Page 12 Line 25: Please give a reference for the WDM6 cloud microphysics scheme.**

Answer: I have added the reference.

Lim, K.S. and S. Hong, 2010: Development of an Effective Double-Moment Cloud Microphysics Scheme with Prognostic Cloud Condensation Nuclei (CCN) for Weather and Climate Models. Mon.Wea.Rev.,138, 1587–1612, https://doi.org/10.1175/2009MWR2968.1

**- Section 3.2.3: The bias in cloud fraction in summer could be related to a too zonal circulation ? Besides, the author could discuss if those biases in cloud cover could have an impact on AOD in the case of the simulations with aerosol-cloud interactions.**

Answer: I have examined the wind field at 850hPa of ERA-Interim and found small differences with that of the control simulation CON in summer. The model does not seem to deviate much from the general circulation seen in the reanalysis dataset.

[Figure]

**- Tables 2, 3 and 4: Please explain which domain is used for these averages. In particular, it should be stated if only land points are considered (as the model is not coupled with ocean, it would be more relevant to show only land grid points).**

Answer: The domain of analysis is given in page 10 line 31. I have added that both land and sea points are considered. I have also added a description in the label of each table: Domain is defined as -10$^o$W, 40$^o$E and 36$^o$N, 70$^o$N.
The lack of coupling definitely has an impact. However I believe it is mainly seen in temperature. Thus I have included a column in table 2 that has the impact (annual) on temperature only over land.

**- Page 15 Line 7: The impact on surface temperature seems to be larger in autumn than in summer, while the AOD is higher in summer. Is there a role of internal variability?**

Answer: Interesting comment. This is true. For Tegen the domain averaged AOD is indeed larger in summer whereas for Macv1 and MACC the mean AOD is the same for summer and autumn. However in summer it is larger over land for all datasets, where we expect a more strong response in temperature. If we use only land points the temperature impact is still larger in autumn  even though not as clearly as was when using both land and sea points. I

do not think that this has to do with the internal variability of the model but with the rather complex nature of aerosol impact. Larger AOD does not necessarily mean larger reduction of radiation since clouds also play a role. The relative change in shortwave radiation is slightly larger in autumn (4 out of 5 ARI simulations) compared to summer. And I see that the impact on temperature seems to be more correlated with the relative changes in shortwave radiation (%) and not so much with the change in $W/m^2$.

I have added (page 21): "Despite the larger AOD in summer, the temperature impact is greater in autumn. This is probably related to the fact that the relative Rsds decrease is slightly larger in autumn (except for ARI_Mv1full). "

**- Page 15 Line 9: "in cases reaching a decrease of 1.5° C" Please give more details on these cases.**
Answer: I have added(page 21): "Cases of such strong reduction are not spatially extended and are seen mainly in summer and autumn within the areas of intense cooling like the Balkans and near of the Black Sea. "

**- Page 15 Lines 10-12: This point should be related to a figure.**
Answer: I referenced figure S3 of the supplement.

**- Page 15 Lines 13-end: The results using the ARI_Mv1urban simulation should be moderated as the absorption of aerosols is not realist in this simulation.**

Answer: I have added: "We must remind here that ARI_Mv1urban is more of an idealized experiment with unrealistically absorbing aerosol."

**- Page 16 Line 5: Could you explain: "Contrary to the ARI group, simulation ACI using the Thompson aerosol-cloud interacting cloud microphysics and accounting for indirect effects only results in a domain averaged temperature increase (0.1 to 0.2 o C) compared to CON for all seasons except autumn" ?**

Answer: ACI just implements the Thompson aerosol-aware scheme compared to CON and no aerosol-radiation interactions. The aerosol-aware scheme presents less cloudiness than Thompson2008 used in CON thus ACI has temperature increase since aerosol do not interact with radiation to decrease it. I understand that describing ACI after the ARI (aerosol-radiation interactions) simulations for each variable might be confusing. I believe that the reorganizing of the Results per aerosol effect helps to make things clearer.

**- Page 16 Line 21: It should be mentioned that the Black Sea is not coupled, which could influence the results on precipitation.**

Answer: Yes it is important to state this especially in this case.
I have added the phrase (page 23): "It should be reminded however that the simulations do not have an ocean-atmosphere coupling, something that can influence the results on precipitation over the Black Sea."

**- Page 17 Lines 2-4: This point should be related to a figure.**
Answer: I have pointed to the figure 4 that shows the clear-sky direct radiative effect.

**- Page 17 Lines 12-13: how is calculated the correlation with AOD in the ARCI simulation (with which AOD) ? The difference in AOD between ARCI and ACI could have an impact ?**
Answer: Nice point. In the above case correlation is calculated between ARCI and ACI with only the ARCI field being used to determine correlation. The fields of ARCI and ACI (number concentrations, AOD is not available in the model for ACI) are very similar. Since ACI has no aerosol-radiation interactions I believe it makes sense to find the correlation between the AOD field that is active and radiation clear-sky radiation change. I would be more worried about the time evolution of the AOD in ARCI (this is stated in the manuscript), something that could make the mean AOD not sufficient for spatial correlation analysis.

**- Page 19 Line 14: it is difficult to draw this conclusion as AOD is not the same in all simulations. I suggest that AOD could be added in Table 4 to discuss this point.**
Answer: The aerosol field in ARCI is the only field that is interacting with radiation. Thus it makes sense to see what happens when I introduce an active I think that comparing ARCI to ACI can give a strong indication as to how the aerosol-radiation interactions are affected when the aerosol-cloud interactions are also present. Especially when we are talking about domain (and possibly seven subdomain) averages over a large period of time (multi-seasonal or multi-annual averages). The AOD field of ARCI is presented. However since ACI is not provided in the model output since ACI lacks aerosol-radiation interactions and AOD is irrelevant. However the concentrations of aerosol, both droplet nucleating (QNWFA) and ice nucleating (QNIFA) are given for each grid point and model level for both ARCI and ACI. If we calculate the total concentration over each grid point the aerosol fields are almost identical. See below for annual averages. Thus I feel confident that the ARCI to ACI comparison can give valuable information regarding the general behavior of aerosol effect over the domain over large time periods. For much smaller spatial and temporal scales the above comparison could be problematic. Finally the idea to add AOD in table 4 (table 2 –new manuscript) is very good and has been implemented.

[Figure]

**- Page 19 Lines 21-24: I don't understand how the author come to this conclusion.**

Answer:  This talks about the simulations ARCI and ACI that use the Thompson aerosol aware microphysics scheme having less cloud amounts and more positive cloud forcing compared to control CON (that uses the Thompson2008 scheme). This point about the aerosol aware scheme is more evident when comparing ACI to CON since the only difference between them is the change in the microphysics scheme. ACI has less cloud fraction amount, more prominently in summer and spring (fig5 original manuscript) and more positive cloud forcing (fig6 and table 4 original manuscript) than CON. Of course these differences cannot be attributed only to aerosol-cloud interactions but mainly to the change of the microphysics scheme.

**- Page 20 Line 1: what is Aer2urban ?**

Answer: My apologies. The naming of the simulations was changed after initial submission due to the editors suggestions and an earlier naming remained in the text. It was changed to ARI_Mv1urban.

**- Page 20 Line 8: Could you explain why the effects are stronger on DNI than on Rsds ?**

Answer: I have added this phrase(page 18): "Since DNI comes only from the direction of the sun, any interaction with aerosol (scattering, absorption) removes radiation amount from this direction. On the other hand in Rsds radiation is reduced only when it absorbed or scattered in an angle that does not reach the surface. Thus the aerosol direct effect is much stronger in DNI."

**- Page 21 Line 5: what is the specific effect of aerosols on diffuse radiation, that could be distinguished from direct radiation ?**

Answer: I have added the phrase (page 18): "Diffuse radiation reaches the surface from all angles except from the direction of the sun (direct radiation). Thus when direct radiation is scattered by aerosol a part of it becomes diffuse radiation and reaches the surface increasing diffuse radiation amount. "

**- Page 23 Lines 11-12. This result involving LWP seems to be important to understand cloud aerosol interactions. Please explain more this process.**

Answer: I believe this is connected to the Thompson aerosol-aware microphysics scheme (TE2014) having in general less cloud fraction amount than the Thompson 2008 microphysics scheme and not so much about the aerosol-cloud interactions. When we change from the Thompson 2008 to the aerosol-aware scheme (e.g. simulations  ACI-CON) we do not just turn on the aerosol-cloud interactions but we introduce a modified microphysics scheme that explicitly predicts aerosol. Thus the impact seen is attributed to the change of the

microphysics scheme and partially also to possible aerosol-cloud interactions. Since I cannot separate these two I do not think I can attribute the decreased LWP to aerosol-cloud interactions. Moreover the aerosol-cloud interactions also rely of the aerosol field that is being produced by the TE2014 scheme. Thus I believe that the change of microphysics scheme presents the main impact. To conclude it seems that the TE2014 has a tendency to produce smaller LWP amounts than the Thompson2008 (at least in our simulations) and this decrease in LWP also leads to decreased cloud fraction amount.

**- Page 23 Lines 19-20. In this kind of semi-direct effect, the internal model variability could be important. The author should mention this point.**

Answer: Yes this is a very interesting comment. We have seen the same behavior (aerosol induced cyclonic anomaly) to some extent in most simulations despite different datasets and aerosol options used. However the impact of different physics parameterizations or/and different initial conditions and period of simulation can be substantial and could substantially modify this effect. I believe that also the real variability of the climate could strongly impact this result, thus longer simulations would be needed to robustly examine this effect.
 I have added to the manuscript (page19): "However the internal model variability as well as the real climate variability could be very important in this kind of complex feedback mechanism. The use of different physics parameterizations, initial conditions and even different time periods can have a large impact and could potentially modify this effect. Therefore it would be interesting to see whether these results are modified in a large physics ensemble simulating a more prolonged time period. "

**- Page 26 Lines 26-27. I don't understand how "the introduction of aerosol-radiation interactions" could lead to "more transparent clouds" ?**

Answer: (This is actually at page 26 lines 6-7). I do understand that the way it is written it is not just confusing but it is also wrong to directly link aerosol-radiation interactions to more transparent clouds (cloud albedo). What is meant is that when aerosol-radiation interactions and/or the Thompson aerosol scheme with aerosol-cloud interactions are implemented the cloud forcing at the surface becomes more positive (more radiation at the surface). This can be due to a change in cloudiness amount due to aerosol semi-direct effect or cloud optical properties due to cloud indirect effect. I have tried to rephrase.

Changes in the manuscript (page20):
From: "Thus, the introduction of aerosol-radiation and/or aerosol-cloud interactions leads to more transparent clouds enabling larger amounts of radiation reaching the surface."

To: "Thus, the introduction of aerosol-radiation and/or aerosol-cloud interactions leads to cloudiness enabling larger amounts of radiation reaching the surface. This can happen due to changes in cloudiness amount or in cloud optical properties."

**Other corrections:**
- Abstract line13 and Page 19 line 6: is comprised of (instead of comprises of)
- Page 4 Line 10: an**d** underestimation
- Page 4 Line 30: incorporates aerosol**s**

- Page 10 Line 15: The MACC reanalysis is **in better agreement** with the satellite data.
- Page 10 Line 16: and the **higher** generally higher AOD
- Figure 1: Please keep the same spelling for **MAC-v1**
- Page 12 Line 11: **In particular** northern Europe is …
- Page 13 Line 21: **error** compensation between errors
- Figure 4: Space character is missing after (RE)
- Page 28 Line 2: please define NWP

Answer: All these corrections have been implemented in the manuscript.

---

## Author Response (AR2)

We would like to thank the reviewer for his time and effort reading the manuscript. The comments have been helpful and contribute to the improvement of the manuscript. We tried to properly respond to all the comments and implement the changes in a new manuscript.

Before proceeding to the responses we need to state that Figure 7 (4th row), contained an error in the previous manuscript. The ARCI-ACI row (4th row) was the same as the ARCI-CON row (bottom row). It has now been corrected.

Now we proceed to answer the comments of the reviewer. The original comments are with bold fonts and each response lies below the respective comment.

**1) Figure 2 -- the authors have added this Figure to demonstrate the response effects of the aerosol-radiation and aerosol-cloud interactions. And the Figure does help to understand the how the tropospheric aerosol layer responds in vertical profile, the regional weather model's high resolution resolving the dynamics and mixing more realistically than in global models.**

**However, the graphs currently show AOD-increments at each model level, whereas to analyse "the vertical distribution of AOD", requires to show the vertical profile with units of extinction per unit altitude. Please re-plot Figure 2 with the graph showing the vertical profile in aerosol extinction (per km).**

Answer: This is indeed a very important and accurate comment. The previous figure presented the AOD (total extinction) in each model layer. Because of the different thickness of model layers (increase with altitude) the AOD per layer is larger not near the surface but higher in the troposphere, despite the extinction coefficient being larger near the surface. But this effect is artificial. We have re-plotted Figure 2, now showing the vertical profile in aerosol extinction coefficient ($km^{-1}$).

**2) Aerosol optical property acronyms/symbols need to have corresponding wavelength.**

**Whenever AOD is given in the text, it should be provided with the corresponding wavelength. The same goes in general for single scattering albedo and asymetry parameter, although since in this manuscript these terms are only discussed in general terms, these can remain without a specific wavelength.**

**Aside from where AOD is discussed in general terms, please change all instances of AOD (and extinction when this is introduced) to be given with the wavelength in subscript. For the Figures the wavelength can be specified in the caption, but specifying the wavelength is essential good practice within the aerosol community and is required throughout section 3 and in Table 2. Section 2.3 is OK since the wavelength is introduced within that subsection.**

Answer: It is indeed essential to state in all instances the wavelength of the AOD. This has been added as a subscript in most cases throughout the manuscript, except in few instances where AOD is discussed in general terms. In some instances (page 8, line 9) this information has also been added for the single scattering albedo and asymmetry parameter. This was something missing. Thank you for this comment.

**3) Section 3.1 -- new para of text beginning "We mainly compare..." -- this para is mostly fine and it helps to give the overview text here, -- minor point "use the SEVIRI product to increase robustness" -- re-word to "as an additional test for the model" or similar. The sentence re: winter and autumn needs to be clarified to better explain what you mean. They are obviously very different types of satellite instrument, so please clarify the specifics of what you mean here, or delete the sentence.**

Answer: We have reworded "to increase robustness" to "as an additional test" (page 11, line 22).

We have also deleted the sentence describing the satellite coverage in winter and autumn (page 11, line 22). Both datasets have several missing values in these two seasons but this explanation is beyond the scope of this paper.

**4) Section 3.1 -- new para of text beginning "The fields of the satellite datasets..." -- this is strange wording to begin the para -- you mean "AOD distribution" I guess -- please re-word.**
**This para is an example of the specific values given for AOD without corresponding wavelength. Please add subscript 550 after AOD. "have larger AOD values" -- you need to generalise to "have a systematically higher AOD on average" or similar more scientific statement.**

Answer: We have re-worded "The fields of the satellite datasets..." to "The $AOD_{550}$ spatial distribution of the satellite datasets" (page 11, line 26).

We have also re-worded "have larger AOD values" to have a systematically higher $AOD_{550}$ on average" (page 11,line 28).

We have also added the wave length subscript for the AOD.

**5) Section 3.1 -- The new paragraph beginning "vertical AOD profiles" -- since AOD is implicitly a column-integrated property it does not make sense to plot the vertical profile, unless presenting per unit altitude -- that is the definition of the property "aerosol extinction". It's just a case of re-plotting Figure 2 to show aerosol-extinction (per km) and re-wording this paragraph to refer to "vertical extinction profile" rather than "vertical AOD profile" -- also in the caption. Similarly decrease in extinction at higher altitude not decrease in AOD at higher altitude.**

Answer: As stated before, we have re-plotted Fig.2 to show the aerosol extinction profile and refer to it as "the vertical profile of aerosol extinction coefficient".

**6) Section 3.3.1 -- This new para seems to have lost the initial subsection 3.3 title. The article has italicised subsections "Clear sky radiation at the surface", "Radiation at the surface and RE" and "Total cloud fraction and cloud forcing", and this looks like a minor editing error that these should be labelled 3.3.1, 3.3.2 and 3.3.3, with the "Aerosol-radiation interactions" as the main sub-section 3.3. This is part of what the reviewers were meaning re: improving the readibility/organisation -- and this section seems to have lost focus and remains to be properly organised into distinct sub-sections. That should be easy to do, but until this is done the article will not be publishable.**

Answer: Section 3 labeling has been re-organized as suggested by the reviewer. The main section "Aerosol-radiation interactions" is now labeled as 3.3 and the different sub-sections regarding the impact on different variables are labeled as 3.3.1, 3.3.2,... 3.3.5

The remaining two main sections are re-labeled 3.4 "Aerosol-radiation interactions with aerosol-cloud interactions present" and 3.5 "The Thompson aerosol aware scheme".

**7) Sub-section "Clear-sky direct radiative effect"**

**A new acronym has been introduced here that is not explained -- some glitch here in the editing and not properly checked. See comment 6 above -- the subsections here need to be organised into distinct sub-sections.**

**Also, in the new para beginning "The Tegen climatology" the significance of the decrease**

**in SSA from 1.0 needs to be explained to the reader as an increase in absorption. Also suggest to introduce the term "forcing efficacy" to similarly better communicate the significance of "the decrease of clear sky radiation per unit AOD".**

Answer: An explanation for the CDRE acronym has been added (page 17, line 10). Also the fact that lower SSA leads to more absorbing aerosol has also been added (page 17, line 20) and we think this helps to clearly state this point.

I was not aware of the term "forcing efficacy". Thank you for suggesting this. However I have seen that this term is in some cases used to describe the response in temperature per unit of forcing regarding the forcings of different agents (aerosol, GHGs, ozone etc.). Therefore I am afraid that using this term could confuse the reader and I have retained the phrase "the decrease of clear sky radiation per unit AOD".

**8) Sub-section "Radiation at the surface and RE"**

**There are some obviously terminology typos here that need to be remedied before publication. I can only assume this section has not been reviewed properly by the co-authors, and perhaps under some pressure to submit within some timescale.**

**For example DNI text seems unnecessary and that para is not communicated adequately, as identified by the reviewers.**

**Please re-word this subsection to ensure the article can be published.**

Answer: We have tried to correct all the typos found and implement minor corrections. For example in several cases the term "radiative effect" was written as "radiative forcing".

We do believe that both the DNI and Diffuse texts are important since they showcase the far greater aerosol impact at the components of shortwave radiation, that to the overall shortwave radiation itself. We have tried to better connect these paragraphs with the rest of the text and point their significance (pages 17-18).

**9) Sub-section "Radiation at the surface and RE" -- Excerpt of text beginning "However the internal model variability" needs to be re-ordered -- I think you just mean there is an interaction here that you're identifying that can contribute to the climate variability -- but again, the new text added is not clear.**

Answer: What is meant here is that such a disturbance in the general wind circulation (cyclonic anomaly) is complex and may be highly affected by the actual variability of the climate and also by the internal model variability. This is because we may find this anomaly

in all ARI simulations but it is present with different intensities and the spatial extend of the statistically significant changes varies a lot between simulations. Therefore, the signal is there but it is not extremely robust. A different time period or more prolonged simulation could yield different results. The same is true for a different selection of physics parameterizations or modified initial conditions. We have made minor changes in the original sentence and added a new sentence that helps better connect it to the previous text and clarify its meaning (page 19, lines 24-25). Please let us know if you think it is still unclear and needs any further clarification.

**10) Table 2 caption -- "optival" -- please correct this and use spell checker on the article.**

Answer: My apologies for these spelling mistakes. It has been corrected.

**11) New text added beginning "Cases of such strong reduction are not spatially extended" is poorly worded and needs to be re-worded -- re-word more scientifically to "are limited in spatial extent" or similar. Also this para says "at grid point level" needs more scientific explanation such as "in localised areas".**

Answer: Both suggestions have been implemented. The phrase "Cases of such strong reduction are not spatially extended" has been re-worded to "are limited in spatial extent" (page 21, line 12).

Moreover the phrase "at grid point level" has been re-worded to "in localized areas" (page 21, line 11).

**12) Last para before section 3.3.3 -- again the new text added is poorly worded.**

**You have "In general the decrease of the radiation at the surface leads again to surface cooling". But that's a tautology. What I think you mean is that the aerosol effects cause a net cooling -- which is expected of course. If this text is summarizing, it's OK to keep this but suggest to re-word along the lines of:**

**"As expected, the net aerosol effects (with both ARI and ACI) lead to a decrease in radiation at the surface" or similar -- but the author team should review this in the context of the rest of this subsection which needs to be properly re-organised -- and almost certainly reduced -- to then just make the main points in subsections as in section 3.2**

Answer: This paragraph is now in section 3.4. This whole section has been re-organized and reduced to more clearly present the main points, keeping in mind the suggestions by the reviewer.

**13) Conclusion sections -- again the new text added weakens the article -- this needs to be checked more carefully. You have "at the grid point level" but this is basic scientific explanation -- you mean locally to a particular region or so -- re-word to properly explain what is meant.**

Answer: The phrase "at the grid point level" has been re-worded to "locally".

Some minor changes and corrections have also been implemented throughout the conclusion section to help make it more readable. If there are more specific changes that the reviewer would like to see, please let us know.

*Below we present the revised manuscripts with tracked changes.*

[revised manuscript text omitted]

---

## Author Response (AR3)

Author's response

We would like to thank the editor for his time and effort reading the manuscript. The comments and suggested changes have been helpful and definitely contribute to the improvement of the manuscript. We tried to properly implement the suggested changes in a new manuscript.

The editor's comments and the implemented changes are presented below. The original comments are with bold fonts and each response lies below the respective comment. The new page and line numbering refers to the new revised manuscript with tracked changes.

**1) Page 13, line 6 -- This new text is clumsily worded, please re-word**
**"as we move towards higher altitudes" instead to "with increasing altitude".**

Answer: Change implemented as suggested by the editor (page 13, line8).

**2) Page 13, line 7 -- Also needs to be more scientifically and specifically worded -- replace "considerably smaller values" with "considerably less extinction" or "considerably less aerosol-extinction". And "compared to the the MAC-v1..." with "than the MAC-v1...." -- removes the 2nd "the" and also makes the text easier to read.**

Answer: Change implemented as suggested (page 13, lines8-9).

**3) Page 13, line 8 -- Again, the wording it clunky -- replace "presents the largest" with "has highest" and delete "per units of altitude" -- readers of the manuscript will know that aerosol extinction is defined to be in units of "per unit altitude", so its implicit in the terminology that is the case and doesn't need to be stated here.**

Answer: Change implemented as suggested (page13, line11).

**4) Page 13, line 14 -- insert comma between "aerosol option used" and "the shape of"**

Answer: Change implemented as suggested (page 13, line13).

**5) Page 13, line 15 -- insert comma between "large variability" and "but it also"**

Answer: Change implemented as suggested (page 13, line16).

**6) Page 17, line 14 -- change "decreases by -5 to -8%" to "decreases by 5 to 8%". Since you've said "decreases", you don't need also to have the minus sign in the percentage values.**

Answer: Change implemented as suggested (page17, line8).

**7) Page 17, line 16 -- Since DRE is an established acronym (Direct Radiative Effect), in this case where you are using an type of DRE (the clear-sky DRE), suggest to use lower-case for the additional letter -- i.e. denote clear-sky direct radiative effect as cDRE. It just makes it easier for the reader to scan the text and see the established acronym also identifying the additional letter that specifies the clear-sky. Suggest however also including lower-case "s" (as well as "c") and a hyphen, i.e. make the terminology cs-DRE rather than CDRE. It's just that c-DRE could confuse some readers who might think the "c" is cloudy or so, whereas "cs" is more recognisably clear-sky. The "CDRE" also is potentially making the reader wonder if there is an existing acronym CDR that it's referring to. Overall, it's just much easier then to identify the "DRE" part of the acronym as the primary acronym when you put c-DRE or better cs-DRE. If you agree, please replace all "CDRE" with "cs-DRE".**

Answer: This is a nice suggestion. It probably helps the reader better understand the concept of the acronym and avoid confusion. I have replaced "CDRE" with "cs-DRE" throughout the manuscript.

**8) Page 17, line 17 -- replace "and CON" with "and the control (CON)" -- it's just then a more understandable sentence -- and also the control is not really an experiment so it's different to the other model runs where you give the acronym -- there is only one control run, so you can just say "the control".**

Answer: Change implemented as suggested. I have replaced "and CON" with "and the control (CON)" (page 17, line11).

**9) Page 19, lines 6-7 -- re-word "a part of it transforms into diffuse radiation and therefore increases diffuse radiation amount", replacing the last part with ".. and therefore increases diffuse radiation" or "increases the diffuse radiation component"**

Answer: The phase "increases diffuse radiation amount" has been replaced by "increases the diffuse radiation component" (page18, line31).

**10) Page 19, lines 7-8 -- Improve the way this sentence is written, reduce the current "Thus diffuse radiation at the surface (DIF) is consistently increased in all simulations, with the exception of ARI_Mv1urban due to a general decrease in cloud fraction amount seen in this simulation" to instead say "As expected, this effect causes an increase in diffuse radiation in almost all simulations, the exception being the ARI_Mv1urban simulation which has a large decrease in cloud fraction (see Figure 6)."**

Answer: Change implemented as suggested (page18, lines32-33).

**11) Page 20, line 6 -- In your revisions you've here replaced "could potentially modify this effect" with "could potentially modify this cyclonic anomaly". I suggest here you refer specifically to this being a "cyclonic anomaly effect" -- i.e. add "effect" after "modify this cyclonic anomaly".**

Answer: Change implemented as suggested (page19, line 29). I think indeed it is more complete to refer to it as a "cyclonic anomaly effect".

**12) Page 20, lines 6-7 -- I think the "cyclonic anomaly effect" you are explaining in this subsection is an example of a broader class of "circulation effects" or "aerosol-circulation effects", for example the effects of aerosols on the South Asian monsoon (e.g. Ganguly et al., 2012; Bollasina et al., 2014). With such monsoon systems dominant seasonal influence within the regional's weather and climate, those aerosol-circulation effects are well recognised, but it sounds like here you are identifying there may be another aerosol-circulation effect for Europe in relation to the influence on this cyclonic anomaly. Suggest modifiying this last sentence from "Therefore, it would be interesting..." instead to note this broader context, perhaps something like "The influence of aerosols on the South Asian monsoon is well recognised (Ganguly et al., 2012; Bollasina et al., 2014) and it would be interesting to explore whether this cyclonic anomaly effect might also be an aerosol-circulation effect important for European weather and climate."**

Answer: This is a very nice recommendation that puts the mentioned cyclonic effect into context. The sentence has been modified exactly as suggested, by deleting the phrase starting with "Therefore it would be interesting…" and adding the phrase "The influence of aerosols on the South Asian monsoon is well recognised (Ganguly et al., 2012; Bollasina et al., 2014) and it would be interesting to explore whether this cyclonic anomaly effect might also be an aerosol-circulation effect important for European weather and climate." (page19, lines30-32)

**13) Page 20, line 15 -- replace "which are mostly affected" with "which are most affected".**

Answer: Change implemented as suggested (page20, line5).

**14) Page 20, lines 16-17 -- replace "our aerosol concentrations" with "the specified aerosol concentrations".**

Answer: Change implemented as suggested.

**15) Page 20, line 21 -- replace "the aerosol related change" with "the aerosol-related change".**

Answer: Change implemented as suggested (page20, line7).

**16) Page 24, line 14 -- re-word "where the indirect effect is also present" -- I'm not sure if here you're referring to model simulations that switch on or off the cloud-indirect effects, or if you're drawing a distinction between different regions, where aerosol-cloud interaction effects may be a larger proportion of the overall aerosol radiative effects. Certainly you need to replace "the indirect effect" with "aerosol-cloud interaction effects". You've given the correct aerosol-radiation effect terminology at the start of the sentence (rather than the older "direct effect" terminology) -- and you need also to replace "indirect effect" with "aerosol-cloud interaction effect" here. If you mean to draw a distinction between different regions, you could replace "in an environment" with "in regions where" and replace "is also present" with "are also important". Or if you mean model simulations -- then replace "in an environment" with "in general circulation model simulations" and replace "is also present" with "are also represented".**

Answer: This part of the text refers to model simulations that switch on or off the cloud-indirect effects and not different regions where the aerosol-cloud effect might be stronger. I have replaced "indirect effect" with "aerosol-cloud interaction effects". I have replaced the phrase "in an environment" with "in general circulation model simulations" and also replaced "is also present" with "are also represented". I believe this indeed makes the text easier to understand and avoid confusion.

The new line is (page24, lines1-2): "In general, the behaviour of aerosol-radiation interactions in general circulation model simulations where the aerosol-cloud interaction effects are also represented is quite similar to the implementation of only aerosol-radiation interactions."

**17) Page 24, line 14 -- replace "behavior" (American English) with "behaviour" (British/Commonwealth English).**

Answer: Change implemented as suggested.

**18) Page 30, line 9 -- add comma after "In this study" and replace "we explore the" with "we have explored the" -- it needs to be in past tense in the conclusions.**

Answer: Change implemented as suggested (page29, line9).

**19) Page 31, line 11 -- re-word "Clouds responded (semi-direct effect) by letting more radiation to reach", joining this up with the previous sentence explaining about the effects of absorbing aerosol. Suggest to either delete "(i.e. urban)" or if you mean the actual urban simulation in the paper, state this instead as "(here in the simulations with "urban" aerosol type)" and continue the sentence with a comma, then to continue also explaining it's a reduction in cloud as", with a reducing cloud-fraction response via semi-direct effect, with more radiation reaching...".That then reads much better -- the phrase "by letting" somehow implies it is a choice, when in fact it is simply the way that system responds.**

Answer: The phrase "Clouds responded (semi-direct effect) by letting more radiation to reach the surface..." refers to all the simulations having aerosol-radiation interactions and not only for the absorbing aerosol. Moreover, I believe that the phrase "by letting" is suitable since the results indicate that since there is not a general cloud fraction decrease, it must be a change in cloud optical properties that drives this effect. Thus, clouds do tend to "let" more radiation to reach the surface. I do understand that the text might be a bit confusing and changes have been made to clarify this.

The new modified text (page30, lines 10-15):

"This reduction is twice as large for aerosol of highly absorbing nature (here in the simulation with the "urban" aerosol type). In all simulations enabling aerosol-radiation interactions, clouds responded (semi-direct effect) by letting more radiation to reach the surface (positive change in the cloud radiative effect). This effect must be attributed to changes in the optical properties of clouds since a general decrease of cloud fraction amount is not detected. This positive change in cloud radiative effect considerably counteracts the impact of the cs-DRE by 20 to 60% (2 to 4 W/m$^2$) depending on season."

**20) Page 31, line 15 -- give a citation for the semi-direct effect -- e.g. Johnson et al. (2004), Allen et al. (2019) -- and also use the term "aerosol-cloud semi-direct effect" throughout the paper rather than just "semi-direct effect".**

Answer: I have given a citation for the semi-direct effect but much earlier in the text, in the introduction, where I think is more suitable. The new addition also informs the reader that the "semi-direct effect" is also called the "aerosol-cloud semi-direct effect".

Revised text (page 2, line 10) : "The considerable impact of the aerosol direct and semi-direct effect (also known as aerosol-cloud semi direct effect; Allen et al., 2019) on the climate …"

I would prefer not to adopt the term "aerosol-cloud semi-direct effect" throughout the manuscript. Since the phrase "aerosol-cloud interactions" is used widely in the manuscript, I am concerned that the reader might confuse the term "aerosol-cloud semi-direct effect" and believe that it refers to some sort of aerosol-cloud interaction. I believe that it is better to inform the reader that the term "semi-direct effect" is also called the "aerosol-cloud semi-direct effect" and proceed with the "semi-direct effect" term.

**21) Page 31, line 21 -- replace "domain averaged cooling" with "domain-averaged cooling".**

Answer: Change implemented as suggested (page30, line23).

**22) Page 31, line 26 -- replace "we also show that" with "we have also shown that".**

Answer: Change implemented as suggested (page30, line27).

**23) Page 31, line 31 -- replace "domain averaged precipitation" with "domain-averaged precipitation".**

Answer: Change implemented as suggested (page30, line32).

**24) Page 31, line 33 -- insert commas before "investigating" and and after "interactions".**

Answer: Change implemented as suggested (page30, lines33-34).

**25) Page 31, line 33 -- replace "presented a clear precipitation reduction" with "found a precipitation reduction"**

Answer: Change implemented as suggested (page30, line33).

**26) Page 31, line 34 -- insert commas between "seasons" and "due to" and between "of SST" and "which in turn".**

Answer: Change implemented as suggested (page30, line33).

**27) Page 31, line 34 -- delete "finally".**

Answer: Change implemented as suggested (page30, line33).

**28) Page 32, line 3 -- insert comma after "highly absorbing aerosols".**

Answer: Change implemented as suggested (page31, line3).

**29) Page 32, line 4 -- replace "Concluding, ..." with "Overall, our study finds..."**

Answer: Change implemented as suggested (page31 line4).

*Other changes*

A.  In the introduction (page2, line12) I have added the following phase and citation:

The substantial impact of centrain aerosol species, such as the African dust, to the greater region has also been established (Tsikerdekis et al., 2019).

*Tsikerdekis A., Zanis P., Georgoulias A.K., Alexandri G., Katragkou E., Karacostas C., Solmon F., Direct and semi-direct radiative effect of North African dust in present and future regional climate simulations, Climate Dynamics, 53, 4311-4336, doi:10.1007%2Fs00382-019-04788-z, 2019.*

B.  A better description of the sections of the manuscript (page3, lines8-12)

[revised manuscript text omitted]